



# Eight-year variations in atmospheric radiocesium in Fukushima city

Akira Watanabe[1,2], Mizuo Kajino[3,4,5], Kazuhiko Ninomiya[6,7], Yoshitaka Nagahashi[1], and Atsushi Shinohara[6]

[1]Faculty of Symbiotic Systems Science, Fukushima University, Fukushima, Fukushima 960-1296, Japan
[2]Institute for Climate Change, Fukushima, Fukushima 960-0231, Japan
[3]Meteorological Research Institute (MRI), Japan Meteorological Agency (JMA), Tsukuba, Ibaraki 305-0052, Japan
10 [4]Faculty of Life and Environmental Sciences, University of Tsukuba, Tsukuba, Ibaraki 305-8572, Japan
[5]Institute of Radiation Emergency Medicine (IREM), Hirosaki, Aomori 036-8564, Japan
[6]Graduate School of Science, Osaka University, Toyonaka, Osaka 560-0043, Japan
[7]Intitute for Radiation Sciences, Osaka University, Toyonaka, Osaka 560-0043, Japan

*Correspondence to*: A. Watanabe (watamay1948@yahoo.co.jp), M. Kajino (kajino@mri-jma.go.jp), and K. Ninomiya (ninomiya@rirc.osaka-u.ac.jp)





**Abstract.** After the Fukushima nuclear accident, atmospheric [134]Cs and [137]Cs measurements were taken in Fukushima city for eight years, from March 2011 to March 2019. The surface air concentrations and deposition of radio-Cs were high in winter and low in summer; these trends are the opposite of those observed in a contaminated forest area. The half-lives of [137]Cs in the

concentrations and deposition before 2015 (275 d and 1.11 y) were significantly shorter than those after 2015 (756 d and 4.69 y). The dissolved fractions of precipitation were larger than the particulate fractions before 2015, but the particulate fractions were larger after 2016. The half-lives of [137]Cs in the concentrations and deposition were shorter before 2015, probably because the dissolved radio-Cs was discharged from the local terrestrial ecosystems more rapidly than the particulate radio-Cs. X-ray fluorescence analysis suggested that biotite may have played a key role in the environmental behavior of particulate forms of

radio-Cs after 2014. However, the causal relationship between the seasonal variations in particle size distributions and the possible sources of particles is not yet fully understood. The current study also proposes a method of evaluating the consistency of a numerical model for radio-Cs resuspension and suggests that improvements to the model are necessary.

*Keywords:* Fukushima nuclear accident, long-term observation, radiocesium, atmospheric radioactivity, precipitation

radioactivity



# 1 Introduction

We conducted eight-year measurements of atmospheric [134]Cs and [137]Cs in Fukushima city after the Fukushima Daiichi Nuclear Power Plant (FDNPP) accident that occurred in March 2011 to understand the time variations in and emission sources of [134]Cs and [137]Cs and to propose effective ways to reduce atmospheric radioactivity. Among the various radionuclides released to the environment, radio-Cs is particularly important due to its abundance in terrestrial ecosystems (the impacts of other nuclides were negligibly small 100 days after the accident; Yoshimura et al., 2020), long half-lives (2.06 y for [134]Cs and 30.17 y for [137]Cs), and bioaccumulation (accumulation in muscle tissues, with biological half-lives of 30-150 d; WHO, 2011). Radio-Cs forms aerosols in the air and is therefore efficiently deposited onto the ground surface via precipitation in addition to via dry deposition. Approximately 30 % of the radio-Cs released in March 2011 was deposited onto the ground surface in Japan (the aircraft-measured deposition on the ground was 2.7 PBq for [137]Cs; NRA, 2012, and the most updated estimate of [137]Cs emissions by the Japan Atomic Energy Agency is 10 PBq; Terada et al., 2020). (The activity of [134]Cs in the environment was equivalent to that of [137]Cs in March 2011). Once radio-Cs is deposited onto the ground surface, it circulates within local terrestrial ecosystems, so the discharge from the local environment to downstream or downwind regions may not be substantial (0.02 - 0.3 % y$^{-1}$ to river; Iwagami et al., 2017, approximately 1 % y$^{-1}$ to atmosphere[1]). Thus, long-term monitoring of atmospheric radio-Cs at even one station may allow us to understand the mechanisms of its circulation in the local terrestrial ecosystems, to estimate the external and inhalation exposure risks to the local residents, to propose efficient ways to reduce health risks to the residents, and to assess the effectiveness of decontamination efforts.

To date, a great number of studies have focused on the circulation of radio-Cs in terrestrial ecosystems (Onda et al., 2020). In terms of the long-term monitoring of atmospheric radio-Cs with a focus on resuspension from the ground surface to the atmosphere, several papers have been published. Based on atmospheric measurements taken in the contaminated forest area of the Abukuma Highlands (30 km northwest of the FDNPP) from October 2012 to December 2014, Ochiai et al. (2016) reported that the surface activity concentrations of [137]Cs were higher in summer and lower in winter and that the time variations of the fine-mode (< 1.1 μm in diameter using an impactor) and coarse mode (> 1.1 μm) behaved differently. The coarse-mode fractions were larger in summer, and the fine-mode fractions were larger in winter. Kinase et al. (2018) conducted surface concentration measurements at four locations in the forest area of the Abukuma Highlands from July 2011 to March 2014 and found that the concentrations of [134]Cs and [137]Cs were lower in winter and early spring and higher from late spring to autumn. Their size-resolved measurements with a six-stage cascade impactor showed that the backup filter (< 0.39 μm) activity concentrations were high in winter, consistent with Ochiai et al. (2016). However, Kinase et al. (2018) found through scanning electron microscopy (SEM) that there were significant amounts of soil dust particles in the backup filter; these particles were larger but bounced off the upper impactor stages. Therefore, they concluded that the sizes of radioactive particles were not

---

[1] The annual resuspension rate to the atmosphere was estimated as 0.047 % y$^{-1}$ by Kajino et al. (2016). However, the current study found that the resuspension rate was likely substantially underestimated (see Sect. 3.5 and Fig. 9). A value of approximately 1 % y$^{-1}$ was obtained from improved simulations, but that manuscript is still in preparation.





small but were actually large (coarse-mode particles). In late spring, the surface concentrations were positively correlated with the wind speed, so they concluded that the wind-blown soil particles carried radio-Cs in this season. In the summer and autumn, the concentrations were positively correlated with temperature but negatively correlated with wind speed, so they concluded that the resuspension mechanisms were different in the winter and summer. The SEM analysis revealed that there were more

abundant bioaerosols in summer than in winter. Based on simulations, Kaijno et al. (2016) indicated that the summer peaks in surface concentrations in the Abukuma Highlands could be accounted for by the bioaerosol emissions from forest ecosystems, even though the emission mechanism remains unknown. Igarashi et al. (2019a) further investigated the mechanisms of bioaerosol emissions in forests in summer by using fluorescent optical microscopic observation and high-throughput DNA sequencing techniques. They suggested that the fungal spores that accumulate radio-Cs may be significantly involved in

resuspension in the forest in summer. Kita et al. (2020) suggested that rain induced the emission of radio-Cs associated with fungal spores in the forest in summer. Minami et al. (2020) combined aerosol flux measurements and a multilayer atmosphere-soil-vegetation model and estimated that the bioaerosol emission flux was on the order of $10^{-2}$ µg m$^{-2}$ s$^{-1}$, which could account for the surface concentrations of $^{137}$Cs in the forests in summer (Kajino et al., 2016; Kinase et al., 2018; Igarashi et al., 2019a). Kinase et al. (2018) also showed that there was no enhancement in the $^{137}$Cs concentration associated with forest fire events in

the region. The surface concentration of $^{137}$Cs was not correlated with that of levoglucosan, which is often used as a marker of biomass burning. These results are distinct from those from Chernobyl, where wildfire plays a key role in the migration of radio-Cs associated with the event (Ager et al., 2019; Igarashi et al., 2020). The contributions of additional $^{137}$Cs emissions from the nuclear reactor buildings of FDNPP to the surface concentrations in Japan were negligibly small compared to the resuspensions from the ground surface (Kajino et al., 2016). On the other hand, unintentional emissions in the premises of

FDNPP such as debris removal operations contributed to some observed sporadic peaks (Steinhouse et al., 2015; Kajino et al., 2016), although the impacts of such events might be small in terms of long-term averages and trends.

The current study is distinct from other studies, as it includes long-term comprehensive measurements (time-resolved and size-resolved measurements of surface air activity concentrations together with measurements of dissolved and particulate forms of activity in precipitation) at an urban/rural location in the Fukushima Basin in the vicinity of contaminated forests in

the Abukuma Highlands. The field observation and the simulation methods are described in Sect. 2. Sect. 3 presents the results for the surface concentrations (Sect. 3.1), deposition amounts (Sect. 3.2), size distribution (Sect. 3.3), chemical compositions (Sect. 3.4), comparison with simulations (Sect. 3.5), and comparison with measurements taken outside Fukushima Prefecture (Sect. 3.6). The seasonal variations and possible emission sources are discussed in Sect. 4.1, the impacts of decontamination and natural variations on the differences in trends before and after approximately 2015 are discussed in Sect. 4.2, the reasons

for the substantial deposition amount in January in Fukushima city are discussed in Sect. 4.3, and major findings and future issues are summarized in Sect. 5. The observation data used in the study are provided as a Microsoft Excel file in the Supplement.

**Figure 1:** Map of Fukushima Prefecture and the surrounding prefectures. The locations mentioned in this study and terrestrial elevations are depicted in the map.

## 2 Methods

### 2.1 Sampling site

The observation site, Fukushima University, is located in Fukushima city, located in the northernmost basin (Fukushima basin) in the Nakadori Valley, surrounded by the Ou mountains to the west and the Abukuma Highlands to the east (Fig. 1). The distance of the observation site from the FDNPP is approximately 60 km. The Nakadori Valley was formed by the Abukuma River, which starts in the mountains in Fukushima Prefecture near the border of Tochigi Prefecture and flows northeast through the central parts of Fukushima city to the Pacific Ocean in Miyagi Prefecture. The major radioactive plumes arrived twice in Fukushima city, on March 15 and 20 (plume #3 and #8, as identified by Nakajima et al. (2017), respectively). These plumes were transported over the Abukuma Highlands (where the peaks are mostly lower than 1,000 m) but were blocked by the



higher Ou Mountains (peaks are 1,000 - 2,000 m) and thus transported along the Nakadori Valley (Nakajima et al., 2017). The land surface of Fukushima city was contaminated mainly on the afternoon of March 15 with plume #3. The air dose rate in Fukushima city started to increase at 17:00 local time (LT), associated with the weak rain that started at 13:00 LT, and peaked at 19:30 LT at a value of 24.0 $\mu$Sv h$^{-1}$.

## 2.2 Surface air concentrations

2.2.1 High-volume air sampler, cascade impactor, and radioactivity measurement

The air samples were collected using high-volume air samplers (Kimoto electric Co., Ltd., Model-120SL) placed on the roof of the building at Fukushima University (37.68°N, 140.45°E) at a height of 25 m from ground level. In this study, we carried out two types of air sampling: time-resolved observations and aerosol size-resolved observations. In the former case, aerosol samples were collected on a quartz fiber filter (Tisch Environmental, Inc., TE-QMA-100). The air suction rate of the sampler was 700 L min$^{-1}$. The typical duration of each sample collection was 24 hours, from May 8 to September 2, 2011. Then, we switched to 72 hours of collection until December 27, 2017; after that, 1 week of continuous collection was performed until March 28, 2019. For the latter observations, a cascade impactor system (Shibata Scientific Technology Ltd., HV-RW) was placed into a high-volume air sampler. The air suction rate was 566 L min$^{-1}$. The aerosols were collected separately by diameter on six quartz filters (Kimoto, TE-236). The range of particle sizes in this system was 0.39-0.69, 0.69-1.3, 1.3-2.1, 2.1-4.2, 4.2-10.2, and >10.2 $\mu$m. (Note that the sizes in the manuscript indicate the 50 % cutoff aerodynamic diameters.) Fine particles with a size of <0.39 $\mu$m were captured on a backup filter (Kimoto, TE-230-QZ). The typical sample collection time for the size-resolved observations was three weeks. In types of both observations, activated carbon fiber filters (Toyobo Co., Ltd., KF-1700F 84 mm$\varphi$) were also placed at the exit of the high-volume air samplers to collect gas-state aerosols.

The collected aerosol samples were sealed into polyethylene bags at Fukushima University. After being shaped into definite shapes, the gamma rays from the samples were measured by high-purity germanium detectors (coaxial with 15, 35 and 40 % relative efficiencies, SEIKO EG&G, ORTEC and coaxial with 40 and 60 % relative efficiencies, CANBERRA) connected to a multichannel analyzer system (MCA7600, SEIKO EG&G) at the Radioisotope Research Center, Osaka University. The radioactivities of $^{134}$Cs and $^{137}$Cs were identified at gamma-ray intensities of 605 keV and 662 keV, respectively. The detection efficiencies of the respective detectors for each gamma ray were determined from the same-shape filter samples from standard $^{134}$Cs and $^{137}$Cs solutions obtained from the Japan Radioisotope Association. The typical measurement time of each sample was 1-3 days. Under these conditions, the detection limits of $^{134}$Cs and $^{137}$Cs were approximately $5 \times 10^{-3}$ Bq. The errors in the measured values are derived from the systematic error of geometrical configuration and the standard sample itself in addition to statistical error. All radioactivities determined by our measurements were corrected at mid sampling times.



The radioactivities of both [134]Cs and [137]Cs were identified for most filter samples. The deviation in concentration between [134]Cs and [137]Cs became larger over time due to the relatively short half-life of [134]Cs. According to the radioactive decay correction performed in March 2011, the activity ratios of [134]Cs/[137]Cs were approximately 1. These ratios are consistent with those in other reports related to FDNPP accident, so we concluded that the detected radiocesium originated from FDNPP

accident. During the measurement period, no radioactivity from [134]Cs and [137]Cs was detected from the carbon filters; that is, the component of gaseous radioactive cesium was negligibly small.

### 2.2.2 Impactor/cyclone system

Since the filters for the high-volume air samples were quartz fiber filters, they could not be used for elemental analysis with X-ray fluorescence spectrometry (XRF). For the XRF analysis, we used an impactor/cyclone system (Tokyo Dylec Corp., no-

number special order, 1100 L min$^{-1}$) in which the aerosols were separated by size into < 2.5 μm and > 2.5 μm using an impactor; those < 2.5 μm and >0.1 μm were sampled in glass bottles (As One corp., 2-4999-07) using a 0.1 μm cyclone with sampling intervals of one month from September 2014 to January 2018. Aerosols larger than 2.5 μm were collected on quartz fiber filters in the system. Aerosol samples in glass bottles (0.1 – 2.5 μm) were defined as fine-mode PM (PM$_f$), and those on quartz fiber filters (> 2.5 μm) were defined as coarse-mode PM (PM$_c$). The radioactivities of [134]Cs and [137]Cs in the samples were also

measured in the same manner.

### 2.2.3 Possible artifacts of impactor measurements

Size separation by an impactor is associated with the artifacts caused by bouncing effects. In fact, in cascade impactor measurements, Kinase et al. (2018) observed abundant coarse-mode particles such as mineral dust and bioaerosol particles in the backup filters due to bouncing effects. In the impactor/cyclone system, the glass fiber filters used as an impaction surface

were immersed in silicone oil to prevent particles from bouncing (Okuda et al., 2015). In this study, silicone oil was not used for the cascade impactor but was used in the impactor/cyclone system. However, the long-duration measurements (such as the monthlong measurements) could be associated with the larger particles that rebounded at the impactor and were collected in glass bottles (Okuda et al., 2015).

### 2.3 Deposition (dry plus wet deposition, dissolved and particulate fractions)

The total deposition (dry plus wet deposition or fallout) samples were collected with a precipitation sampler (Miyamoto Riken Ind. Co., Ltd., RS-20) with a funnel diameter of 20 cm. Since a heating device was not installed on the sampler, any snow in the funnel was manually melted in a water bath in winter. The accumulated snow in the funnel never reached the top of the funnel during the whole observation period. A filtration device was installed in the sampler using membrane filters (Advantec, 4-880-03) with a pore size of 0.45 μm. The radioactivities of [134]Cs and [137]Cs in the filtered water stored in the polyethylene

bottle and those on the filters were both measured by high-purity germanium detectors at Osaka University and were defined





as the dissolved and particulate fractions of the deposition, respectively. It should be noted here that this separation does not perfectly differentiate water-soluble and insoluble radio-Cs. The clogging of the pores of the membrane filter can occur during filtration. The measured total (dissolved plus particulate) deposition amounts were compared with those measured by the official method at the Fukushima Prefecture Nuclear Power Center (Fig. 1), which is located 6.5 km north-northwest of

Fukushima University. Our method was found to be consistent with their official method: the correlation coefficient $R$ was 0.81, with a slope of 1.158 (the values from Fukushima University were larger by 16 %). Differences in locations and sampling intervals (daily at the Fukushima Prefecture Nuclear Power Center; monthly at Fukushima University) could also have contributed to the differences in the measured values at the two sites.

## 2.4 X-ray fluorescence analysis (aerosols, deposition, and river sediments)

X-ray fluorescence (XRF) analysis was carried out by using a RIX1000 (Rigaku Corp.) at Fukushima University. The measurement setup recommended by the manufacturer was used for the XRF. The major and trace element contents were analyzed by the fundamental parameter method and calibration curve method, respectively (Takase and Nagahashi, 2007). Measurements were conducted for $PM_f$ (see Sect. 2.2.2), the particulate fractions of precipitation (see Sect. 2.3), and the river sediments. River sediments were collected at 15 sites upstream and downstream of Fukushima city in the Abukuma River and

its tributaries in 2010. Samples were taken from the gravel layer of the lower terrace at 5 sites, from alluvial fan deposits at 1 site, and from current riverbed sediments at 9 sites. The dried sediment samples were sieved and divided into two grain size groups: particles smaller than 180 μm (defined as fine sediment particles) and particles 180 μm - 2 mm (coarse sediment particles).

## 2.5 Numerical simulation and validation data

Kajino et al. (2016) used a Lagrangian model (LM) to simulate the atmospheric dispersion and deposition of $^{137}Cs$ resuspended from bare soil and forest ecosystems from January to December 2013. Since the resuspension fluxes and size distributions were unknown, they adjusted the flux from bare soil (forest ecosystems) so that the simulated surface concentrations matched those measured in Namie (Tsushima) (Namie High School Tsushima Campus, 37.56°N, 140.77°E, 30 km northwest of the FDNPP) (Fig. 1) in the winter (summer) of 2013, and they adjusted the dry and wet deposition parameters (reflecting the size

distributions and hygroscopicity) so that the simulated total (dry plus wet) deposition over land in March 2011 matched those measured by the aircraft measurements (NRA, 2012). Thus, note that the size distribution of the simulation was assumed to have submicron size ranges that were consistent with those of the primary emissions (the direct emissions associated with the FDNPP accident in March 2011) but that may not be applicable for resuspension events; the carrier aerosols are presumed to be soil dust or bioaerosols, which are usually larger than the submicron size range. Kajino et al. (2016) concluded that their

simulations are likely reliable because the simulated differences between the surface concentrations in the contaminated area (or emission source area) (i.e., Tsushima) and those in the downwind area (Meteorological Research Institute (MRI), Tsukuba





city, 36.06°N, 140.13°E, 170 km southwest of the FDNPP) (Fig. 1) were consistent with the observed differences at the two locations.

However, Kajino et al. (2016) used only surface concentration measurements to validate the simulations. The current study also used concentration and deposition measurements from Fukushima University for model validation. The previous study compared only the two locations in the contaminated forest areas and in the downwind urban/rural regions; the current study includes an additional location in the urban/rural region near the contaminated forest of the Abukuma Highlands (60 km northwest of the FDNPP).

## 3 Results

### 3.1 Surface air concentrations

Figure 2 shows the time variations in the atmospheric radioactivity concentrations of $^{137}$Cs from May 2011 to March 2019. Just after the accident, the $^{137}$Cs concentrations were higher than 0.01 Bq/m$^3$, and the maximum concentration of 0.0169 Bq/m$^3$ was detected on May 23, 2011. The concentration quickly decreased to a level of $10^{-4}$ Bq/m$^3$, and the minimum concentration of $4.05 \times 10^{-6}$ Bq/m$^3$ was obtained on December 5, 2018. By taking the annual averaged value, the decreasing tendency in the atmospheric concentration could be expressed as $Y = 0.0418X^{-0.476}$, where $Y$ indicates the annual mean $^{137}$Cs concentration and $X$ means the number of years elapsed. The coefficient of determination, $R^2$, is 0.993. This demonstrates that the surface concentration decreased exponentially and halved in approximately 4 years; thus, the decrease rate was higher than the rate of radioactive decay of $^{137}$Cs.

It is remarkable that the decreasing trends in the earlier stage and the later stage were different. The regression lines of the raw data time intervals for the whole period (red; May 2011 - March 2019), the earlier stage (blue; May 2011 - December 2014), and the later stage (green; January 2015 - March 2019) are shown in Fig. 2, with the half-life ($T_h$) in days and the decrease rate ($R_d$) in % y$^{-1}$. The decreasing trend ($T_h$ = 275 d, $R_d$ = 92.0 % y$^{-1}$) of the earlier stage is approximately three times faster than that of the later stage ($T_h$ = 756 d, $R_d$ = 33.5 % y$^{-1}$). It is shown later in Fig. 3 in Sect. 3.2 and discussed in Sect. 4.2, but this could be related to the relative abundance of particulate and dissolved fractions of radio-Cs in the environment. The dissolved fractions of radio-Cs may discharge faster than the particulate fractions from contaminated environments, such as soils and plants. The relative abundance of the dissolved fractions was larger in the earlier stage than in the later stage such that the decreasing trend in the surface air concentration was faster than that in the later stage. In addition to the natural variability, decontamination work, which was completed by March 2018 in Fukushima city and the surrounding municipalities, may also have contributed to the difference in the decrease rates; this possibility is also discussed in Sect. 4.2.


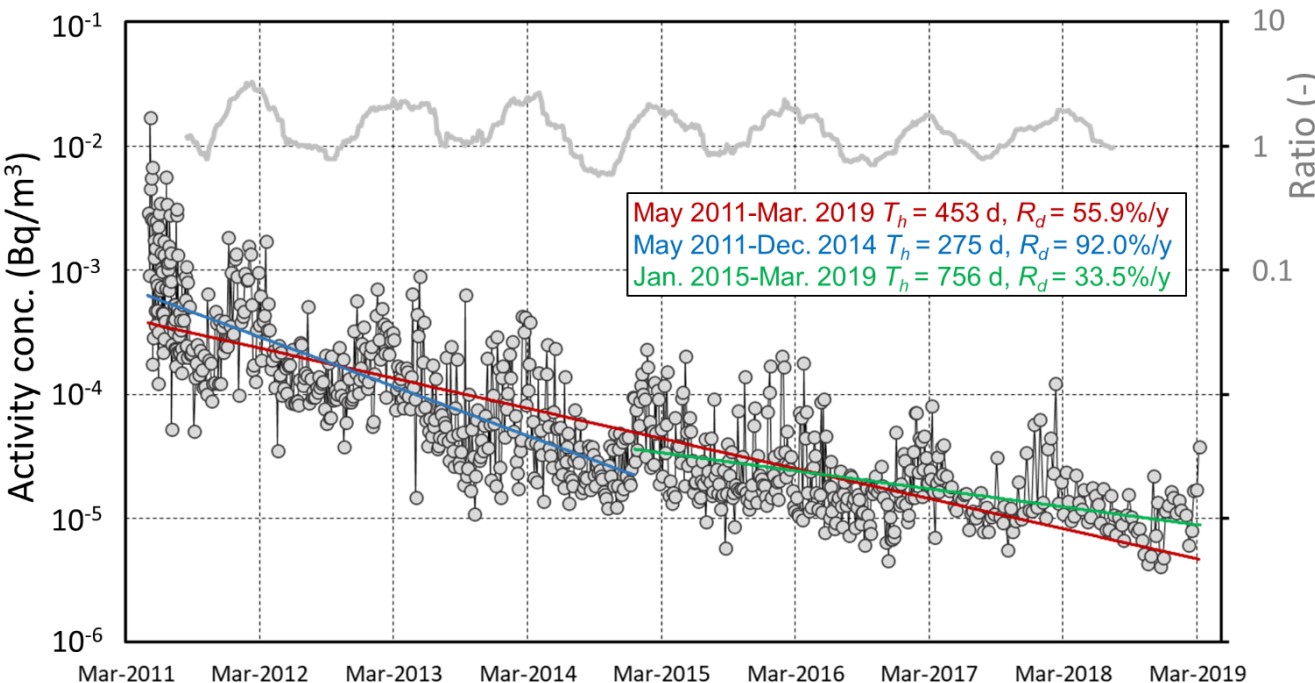

**Figure 2:** Time series of surface air activity concentrations of $^{137}$Cs on the left axis. The red, blue, and green lines indicate the regression lines of the whole period, before 2015, and after 2015, respectively. The half-lives ($T_h$) and decay rates ($R_d$) are also depicted. The gray line indicates the ratio of the running mean of 20 data points (an approximately monthly cycle) to the running mean of 160 data points (an approximately annual cycle) on the right axis to show its seasonal variation.

It is also interesting that our data show different seasonal variations from those measured in Tsushima by Ochiai et al. (2016) and Kinase et al. (2018). The levels in their studies were high in summer and low in winter, but as depicted in the gray line in Fig. 2, the concentration rose starting in October, with maxima in the spring season around March and minima in the summer. The maxima in the spring are approximately one order of magnitude larger than the minima in the summer. The measurements of their studies were conducted in high-dose areas in the mountain forest (approximately 400 m above sea level (a.s.l.)), and the high-volume samplers were set near the ground surface. In contrast, the current air sampling was conducted in a relatively low-dose area (10 times lower than that in Tsushima) located in an urban/rural region on a hill (approximately 200 m a.s.l.) at the southern end of the Fukushima Basin. The heights of the samples were 25 m from the ground surface. Such geographical and altitude differences could have caused these differences.





## 3.2 Deposition amounts

Figure 3 shows the monthly cumulative deposition of $^{137}$Cs from March 2011 to March 2019. The monthly deposition amount peaked in March 2011 as 202200 Bq m$^{-2}$, decreased to 1 % of the initial amount after one year, and decreased to an order of 1 Bq m$^{-2}$ after eight years. It also showed seasonal variation and was high from winter to spring. Nevertheless, the current level

is from two to three orders of magnitude larger than that before the Fukushima nuclear accident. The monthly trend is expressed as $Y = 48232X^{-1.944}$, where $Y$ indicates the monthly cumulative $^{137}$Cs deposition and $X$ means the number of months elapsed. $R^2$ is 0.697.

The decrease rates of deposition ($T_h = 1.11 - 4.69$ y) were generally slower than those of the surface concentrations ($T_h = 275 - 756$ d). It is hard to identify the reason for this phenomenon. A perfect simulation could answer this question, but

high uncertainties in atmospheric deposition modeling and land surface modeling inhibit a perfect understanding of these long-term circulations of radio-Cs in the environment. It is safe to presume here that the decreasing trends in deposition and surface concentrations are different because the contributions of major emission sources to deposition and surface concentrations are different. If the dominant source of the surface concentration is near (far from) the observation site and that for of deposition is far from (near) the site, the faster decrease rate in concentration is due to the faster (slower) reduction rate in the nearby

sources of emissions than in the far sources.

There is also a distinct difference in the decreasing trends before and after 2015. In addition to the effect of decontamination work, as previously discussed in Sect. 3.1, the relative abundances of the dissolved and particulate fractions of $^{137}$Cs could be a part of the reason. The particulate fraction made up 72.6 % of the deposition of March 2011, which is presumed to be largely influenced by primary emissions. Here, it is interesting to note that most primary radio-Cs emissions

are thought to be composed of water-soluble submicron aerosol particles (e.g., Kaneyasu et al., 2012 and almost all numerical simulations afterwards, such as Sato et al., 2020), while water-insoluble Cs-bearing microparticles (CsMP; Adachi et al., 2013, Igarashi et al., 2019b) may contribute somewhat to primary emissions (Ikehara et al., 2020, Kajino et al., 2021). If the primary radio-Cs in aerosols were 100 % in water-soluble forms, the particulate fraction should have made up 0% of the precipitation in March 2011 (although, some of the water-soluble Cs could have converted to a water-insoluble form through adsorption to

soil particles accumulated on the membrane filter during filtration). After April 2011, as the contributions of resuspension were thought to be dominant, the dissolved fractions became larger. The ratio varied, but the dissolved fractions were generally higher before 2016, and the particulate fractions became dominant after 2016. There seemed to be a regime change in the physicochemical properties of radio-Cs circulating in the environment in the area in approximately 2015 and 2016, which could have changed the decreasing trends of both the surface concentrations and deposition before and after 2015. This result

is consistent with the finding of Manaka et al. (2016), who reported that the exchangeable proportions of radio-Cs rapidly decreased in forest soils from two to four years after the accident, i.e., from 2013 - 2015.





**Figure 3:** (Top) Time series of $^{137}$Cs activity deposition. The red, blue, and green lines indicate the regression lines of the whole period, before 2015, and after 2015, respectively. The half-lives (Th) and decay rates (Rd) are also depicted. (Bottom) Time series of particulate and dissolved forms of $^{137}$Cs deposition on the left axis and the ratio of particulate to dissolved $^{137}$Cs on the right axis.





The seasonal variations in particulate and dissolved [137]Cs were slightly different from each other and different from those of the surface concentration. The surface concentration peaked in March in almost all years, and the total deposition peaked in January. The peaks of the total deposition in January coincided with those of the dissolved [137]Cs before 2016, but the peaks of the dissolved [137]Cs became unclear afterwards. The peaks of particulate [137]Cs occurred in March before 2016, which coincided with those of the surface concentrations. After 2016, there were no clear seasonal variations in particulate [137]Cs. There are clear and different seasonal variations in the surface concentration and deposition. However, at the current stage, we have no knowledge of or numerical tools to reveal the hidden mechanisms underlying these variations.

### 3.3 Size distributions

Figure 4 shows the time series of the seasonal mean atmospheric radioactivity concentrations of [137]Cs obtained from the cascade impactor measurements. The sampling interval for the cascade impactor measurements was three weeks. The seasonal means included a sampling period if any part of the sampling period was included in the season. For example, the raw data from the sampling period from February to March contributed to the averages of both DJF (December, January, and February, i.e., winter) and MAM (March, April, and May, i.e., spring). The seasonal mean total (all sizes) concentrations of cascade impactor measurements during the sampling period agreed well with those of the time-resolved observations (Fig. 2), with $R^2$ = 0.93. The same seasonal variation discussed for the time-resolved observations (Sect. 3.1) was also observed; the atmospheric [137]Cs concentration was relatively high in DJF and MAM compared to that in JJA (June, July, and August, i.e., summer) and SON (September, October, and November, i.e., autumn).

The most dominant size range in activity was the backup filter (< 0.39 μm, or rebounded particles such as soil dust and bioaerosols; Kinase et al., 2018), and its seasonal variation agreed well with that of the total particle concentration (high in DJF and MAM). On the other hand, the second largest contribution was made by the size range of 4.2-10.2 μm, which showed the opposite seasonal variation and was relatively high in JJA and SON. The seasonal variations in the largest particle fraction, larger than 10.2 μm, are interesting. The trend appears to be synchronized with that of the backup filter particles (high in DJF and MAM), but the opposite trend was observed in 2016 and 2017 (high in JJA). The contributions of other fractions, i.e., 0.49-4.2 μm, were small in the measured period. Even though the contributions were small, the seasonal trend of 0.39-0.69 μm was similar to that of the backup filter particles, but that of 1.3-2.1 μm was similar to that of 4.2-10.2 μm. The current measurement indicates that the dominant particles and their sizes may be distinct depending on the season. The decrease rates of each size were different before and after approximately 2015, as discussed in Sects. 3.1 and 3.2, but the size distribution of the surface activity did not change substantially before and after approximately 2015.





**Figure 4:** (Top) Time series of seasonal mean size-resolved surface activity concentrations of $^{137}$Cs and (bottom) their relative fractions.



**Figure 5:** (Top) Time series of surface activity concentrations of $^{137}$Cs in PM$_f$ (0.1-2.5 µm) and PM$_c$ (>2.5 µm) collected by the impactor/cyclone system and those of the backup filter of the cascade impactor. (Bottom) Correlation coefficients of temporal variations among seasonal mean $^{137}$Cs activity concentrations of different sizes measured by the impactor/cyclone and the cascade impactor. Correlation coefficients higher than approximately 0.4 and lower than approximately -0.4 are colored blue and orange, respectively.



Cascade impactor sampling is associated with the bouncing effect, whereas filters for the impactor/cyclone system were immersed in silicone oil to prevent the bouncing effect. Thus, compared the cascade impactor and the impactor/cyclone measurement data, as shown in Fig. 5. The top panel of Fig. 5 shows the data with the same measurement time intervals (three weeks for the cascade impactor data and one month for the impactor/cyclone data). The surface activity concentrations of $^{137}$Cs

in the backup filters were well correlated with those of PM$_f$. No remarkable seasonality was observed in PM$_c$, but some enhancements were observed in JJA in 2015 and SON in 2016.

The bottom panel of Fig. 5 shows the correlation coefficients among the seasonal mean size-resolved data from the cascade impactor and impactor/cyclone measurements. If we assume that the bouncing effect on the impactor/cyclone measurements was negligible, the cascade impactor data and the impactor/cyclone data were consistent. There was a positive

correlation between PM$_f$ and the backup filter data. There were also positive correlations between PM$_c$ and the 1.3-2.1 µm and 4.2-10.2 µm data. There was a negative correlation between PM$_c$ and PM$_f$. We can assume that fine-mode particles are the dominant carriers of $^{137}$Cs in winter and spring and that coarse-mode particles are the dominant carriers of $^{137}$Cs in summer and autumn. However, there was also a contradiction in the data. There were low or negative correlation coefficients between the backup filter data and the cascade impactor data at smaller size ranges, such as 0.39-0.69, 0.69-1.3, and 1.3-2.1 µm, but

the backup filter data were positively correlated with the impactor data for > 10.2 µm. It appears that bouncing effect occurred; particles larger than 10.2 µm bounced in the latter stages and were captured in the backup filter. However, as previously discussed, the behaviors of the >10.2 µm-particle data were not consistent in time, i.e., they were generally high in DJF and MAM and were high in JJA in 2016 and 2017 (Fig. 4). Kinase et al. (2018) and Igarashi et al. (2019a) considered that the dominant carriers of resuspended $^{137}$Cs were coarse-mode particles such as soil dust and bioaerosols. Ochiai et al. (2016)

conducted two-stage impactor sampling and measured the surface activity concentrations of $^{134}$Cs and $^{137}$Cs above and below 1.1 µm from 2012 to 2014. They showed that the contributions of coarse-mode particles (> 1.1 µm) were dominant, with maxima in summer. The contributions of the fine-mode particles (< 1.1 µm) were much smaller, and no significant seasonal variations were found. All of their measurement sites were surrounded by contaminated forests in the Abukuma Highlands (Tsushima and the nearby sites), so the sampling sites were different from those in our study. Such larger particles may have

contributed to the backup filter data in the current measurements; however, based on the fact that the backup filter data were positively correlated with PM$_f$ and not with PM$_c$, fine-mode particles (< 2.5 µm) should also play a key role in determining the surface air concentrations in Fukushima city.

On the other hand, if we assume that the bouncing effect is also significant in the impactor/cyclone system due to the long sampling duration, as suggested by Okuda et al. (2015), the positive correlation between the backup filter particles and

PM$_f$ was simply due to the bouncing effects of the larger particles in both systems.




Even though the emission sources of the dominant particles collected by the size-resolved measurements could not be identified in this study, the possible aerosol sources that would explain the differences in size and seasonality of the two locations are discussed later, in Sect. 4.1.

**3.4 Chemical characterizations of particles in the air, rainfall, and river sediments**

Figure 6 shows the relative abundance of the XRF-measured atomic number concentrations of elements in the $PM_f$ monthly sample from September 2014 to January 2018. Among the 15 detected species, $PM_f$ was mainly composed of $SiO_2$, $Al_2O_3$, and $SO_3$. The fractions of $SiO_2$ show clear seasonal variations and were higher around May. The seasonal variations in $Al_2O_3$ and $SO_3$ are the opposite of that of $SiO_2$. A positive temporal correlation was obtained between the $^{137}Cs$ in $PM_f$ and $SiO_2$ ($R =$ 0.30). Negative correlations were obtained for $Al_2O_3$ and $SO_3$, with correlation coefficients of -0.36 and -0.35, respectively. Note that these results do not prove that the $SiO_2$-bearing aerosols are the carriers of resuspended $^{137}Cs$, but we can safely conclude that the origins of $SiO_2$ and $^{137}Cs$ may be close to each other (i.e., that both come from the same source or the same area/direction).

Figure 7 shows comparisons of the relative abundance of the periodic mean XRF measured atomic number concentrations in different samples, fine sediment particles, coarse sediment particles, $PM_f$, and particulate fractions of precipitation. The $PM_f$ and precipitation data over the same period, from October 2014 and December 2012, were averaged. The sediment samples were collected in 2010. The 10 species that were common to all samples are shown in Fig. 7. The correlation coefficients for the compositions among samples are above 0.9, showing that the samples have similar origins. The features of the $PM_f$ composition were distinct from the others. $PM_f$ included $SO_3$ (17.8 %) and Cl (2.65 %), while the others did not.

Weathered biotite is abundant in the soil in Fukushima and absorbs radio-Cs efficiently (Kogure et al., 2019). The compositional correlation coefficients between the weathered biotite (Takase, 2020) and the four samples were high, at 0.73 to 0.87. However, when the two major components $SiO_2$ and $Al_2O_3$ were excluded, the compositional correlation coefficients changed significantly. The eight compositional correlation coefficients between the fine and coarse sediment particles were 0.98, but those between the sediments and $PM_f$ were 0.01 and 0.19 for the fine and coarse sediment particles, respectively. The eight compositional correlation coefficients for the particulate fractions of precipitation were moderate, at 0.36, 0.44, and 0.45 for fine sediment particles, coarse sediment particles, and $PM_f$, respectively. The eight compositional correlation coefficients for weathered biotite were 0.76, 0.71, 0.50, and -0.14 for fine sediment particles, coarse sediment particles, particulate fractions of precipitation, and $PM_f$, respectively.


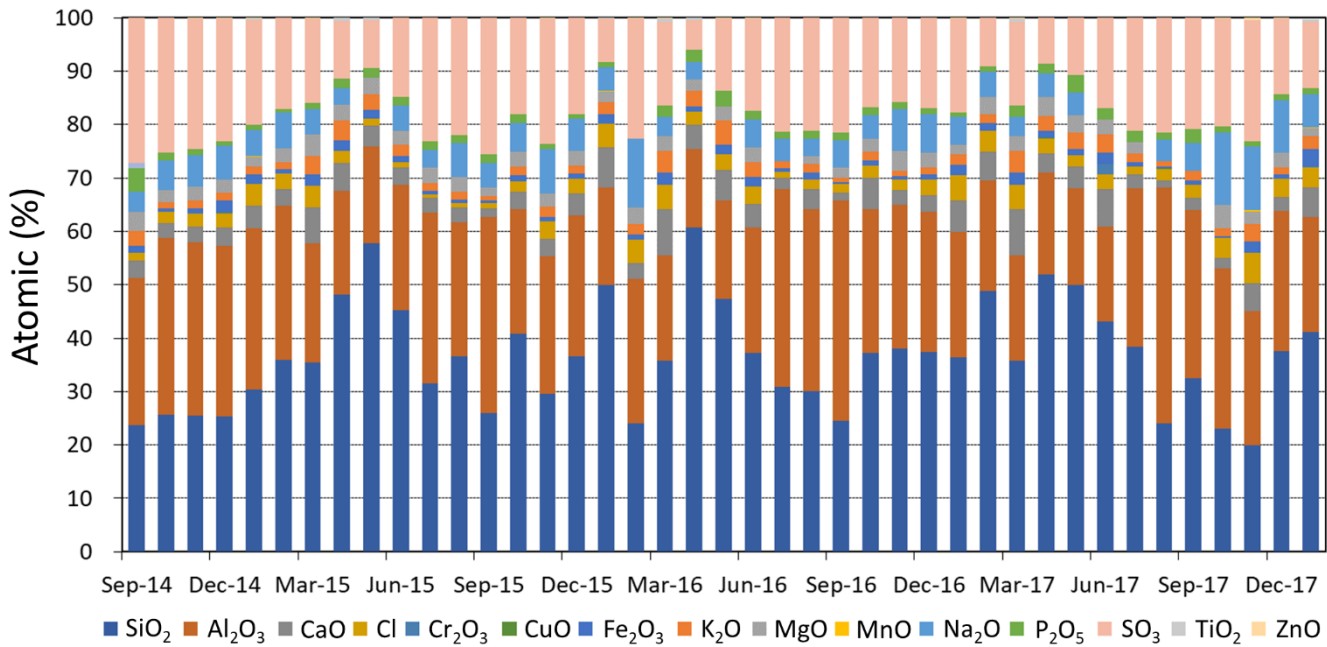

**Figure 6:** Temporal variations in the chemical composition of $PM_f$ as measured by XRF.

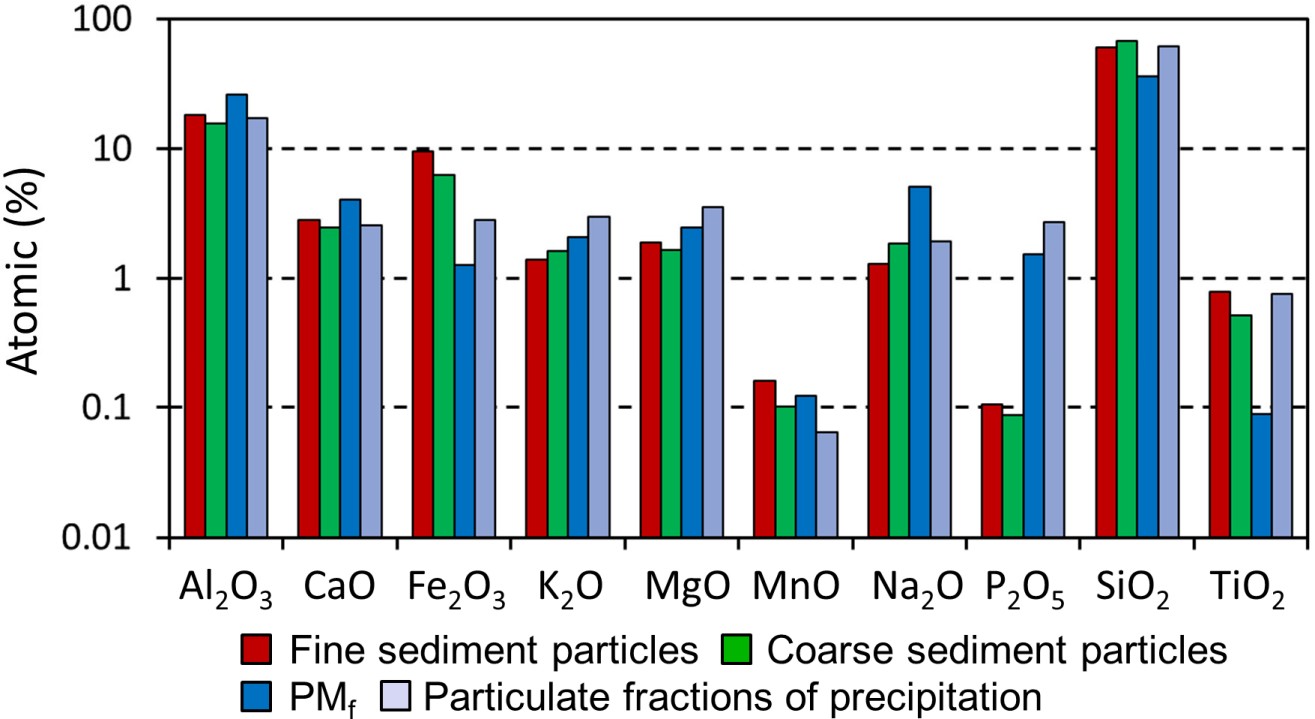

5 **Figure 7:** Periodic mean chemical compositions of fine particles and coarse particles in sediments, $PM_f$, and the precipitation filter (the particulate fraction of precipitation) measured by XRF.





The findings from the current section are summarized as follows. The mean compositions of both fine and coarse sediment particles are similar to those of biotite, which absorbs radio-Cs efficiently. The similar composition feature was observed for the particulate fractions of precipitation. The composition of $PM_f$ was slightly different from those of the other samples, but the $^{137}Cs$ concentrations in $PM_f$ become larger when the relative fractions of $SiO_2$, the major component of biotite, increased. Thus, biotite may have played a key role in the environmental behavior of radio-Cs in Fukushima city since September 2014. However, the major carriers of radio-Cs before September 2014 and those in the dissolved fractions in precipitation are still unknown.

**3.5 Comparison with the simulation results and climatological deposition velocity analysis**

In Fig. 8, the surface concentrations of $^{137}Cs$ in 2013 simulated by Kajino et al. (2016) are compared with the time-resolved observation data (Fig. 2). Kajino et al. (2016) included $^{137}Cs$ resuspended from bare soil, $^{137}Cs$ resuspended from forest ecosystems, and additional $^{137}Cs$ emissions from the FDNPP. The additional $^{137}Cs$ emissions were negligibly small, the concentrations of which in East Japan were two to three orders of magnitude smaller than those from the two sources. Therefore, they are not depicted in the figure. The simulation was successful in explaining the magnitude and seasonal variations in surface concentrations at Tsushima and the MRI, but the simulation at Fukushima city disagreed with the observations. The simulation showed an enhancement of $^{137}Cs$ from forests in the summer, but that was not detected in the observations. The observed magnitude and seasonal trends are rather similar to those simulated for $^{137}Cs$ from soil dust.

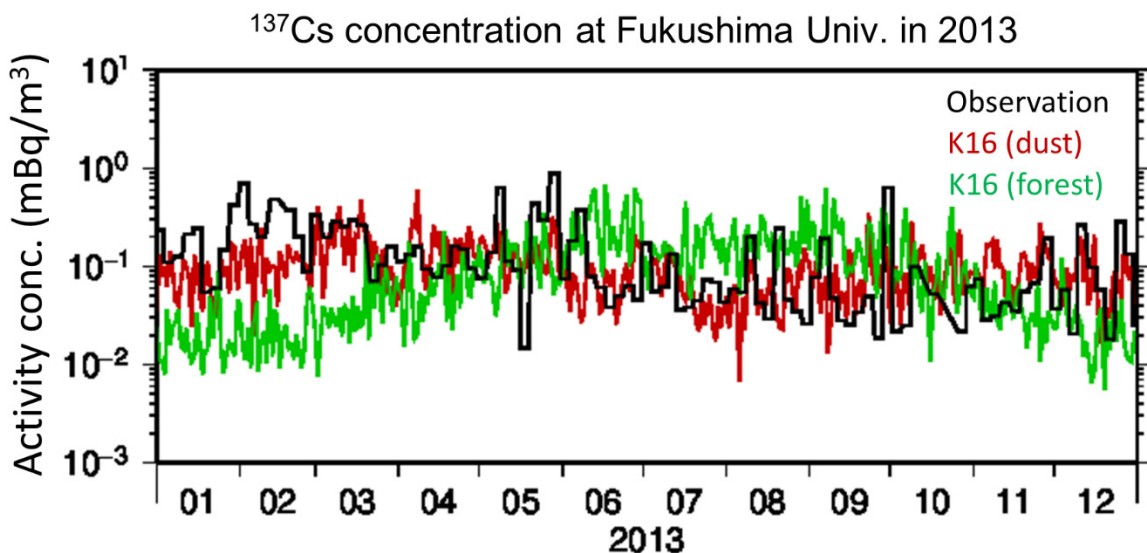

**Figure 8:** Time series of (black) measured surface activity concentrations of $^{137}Cs$ and those simulated (by Kajino et al., 2016; K16) considering different emission sources, (red) mineral dust from bare soil and (green) aerosols emitted from forest ecosystems.





Kajino et al. (2016) used only the observed surface concentrations to estimate the regional budget of resuspended [137]Cs in the air, but we used the observed deposition to evaluate the model, as shown in Fig. 9. Suppose there is a simple nonlinear relationship between the deposition ($D$) and surface concentration ($C$):

$$D = aC^b, \qquad (1)$$

where $a$ represents a removal rate and $b$ represents nonlinearity, such as spatial and temporal variabilities. If one can take a
long-term average of $D$ and $C$, Eq. (1) may hold. Eq. (1) is reformulated as

$$log(D) = b \, log(C) + log(a). \qquad (2)$$

The log-log scatter plot between the monthly mean surface concentrations and monthly cumulative deposition of observed (purple) and simulated (orange) [137]Cs are depicted in the left panel of Fig. 9. The coefficient of determination of the observation was 0.678, with a risk factor for < 0.1 %. Eq. (1) holds for the monthly mean resuspended [137]Cs at Fukushima University. As seen in Eq. (2), the intercept of the Y-axis indicates the removal rate $a$. $log(a)$ is dimensionless, but if $b$ is close to one, the unit
of $a$ can be m s[-1]. From Fig. 9, $b$ of observation is close to one. Therefore, the ratios of the monthly deposition amounts to the monthly mean surface concentrations are referred to as the climatological deposition velocity (m s[-1]). Time series of the climatological deposition velocity are presented in the right panel of Fig. 9.

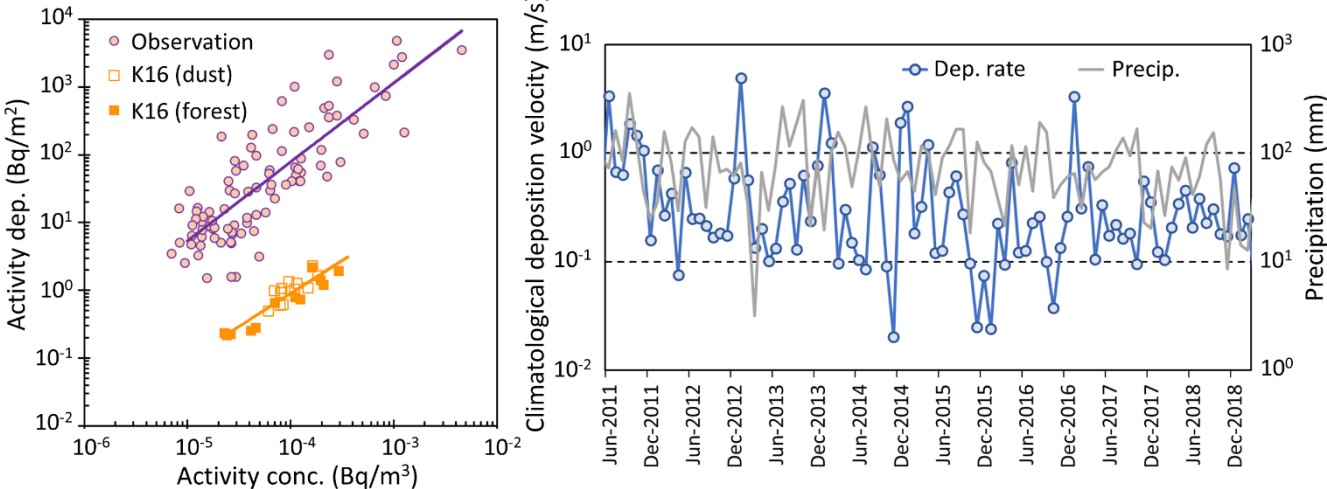

**Figure 9:** (Left) Scatter diagram of (purple circles) observed surface concentrations and deposition of [137]Cs and those simulated
(by Kajino et al., 2016; K16) considering different emission sources, (orange open squares) mineral dust from bare soil and (orange close squares) aerosols emitted from forest ecosystems. The purple and orange lines indicate the regression lines of the observed data and the simulated (both dust and forest) data. (Right) Time series of (blue) climatological deposition velocity on the left axis and (gray) precipitation amounts on the right axis.





The left panel of Fig. 9 clearly shows that the removal rate ($a$) used in Kajino et al. (2016) can be underestimated by one to two orders of magnitude. The deposition velocities used in Kajino et al. (2016) were estimated from the observation of $^{137}$Cs in March 2011, which was supposed to be mainly composed of submicron water-soluble particles. However, the current study and the series of previous studies regarding resuspended $^{137}$Cs indicated that the host particles of $^{137}$Cs could be

substantially larger (e.g., soil and bioaerosols). This may be the reason for the overestimation of simulated $^{137}$Cs from forests in the summer in Fukushima city. If the deposition velocities of the model increased by one to two orders of magnitude, the transport of $^{137}$Cs from the contaminated forest to Fukushima city in summer may decrease such that the simulated surface concentration in Fukushima city agrees with the observation. Certainly, their simulated regional budget needs to be reassessed using the realistic deposition velocities indicated in the current study.

The observed climatological deposition velocity varied by more than one order of magnitude over time. There are two main deposition mechanisms: dry deposition and wet deposition. Wet deposition is associated with precipitation. The variations in the climatological deposition rate seem to agree with the observed precipitation, but almost no correlation was observed ($R \sim 0.10$). The mean climatological deposition velocity was $5.3 \times 10^{-1}$ m s$^{-1}$, and the peak values occurred in January. The maximum value was 4.9 m s$^{-1}$ in January 2013, when the monthly precipitation was not very high (81.2 mm). Possible

reasons for these peaks in January are discussed later, in Sect. 4.3. The typical order of the dry deposition velocity of supermicron (1–10 μm in diameter) particles is approximately $10^{-3}$–$10^{-2}$ m s$^{-1}$ (e.g., Petroff and Zhang, 2010), which is substantially lower than the values in our climatological deposition velocity analysis. Certainly, the magnitudes of the instant deposition velocity and our climatological deposition velocity are not directly comparable, but it seems that wet deposition plays an important role in the removal of resuspended $^{137}$Cs-bearing atmospheric aerosols.

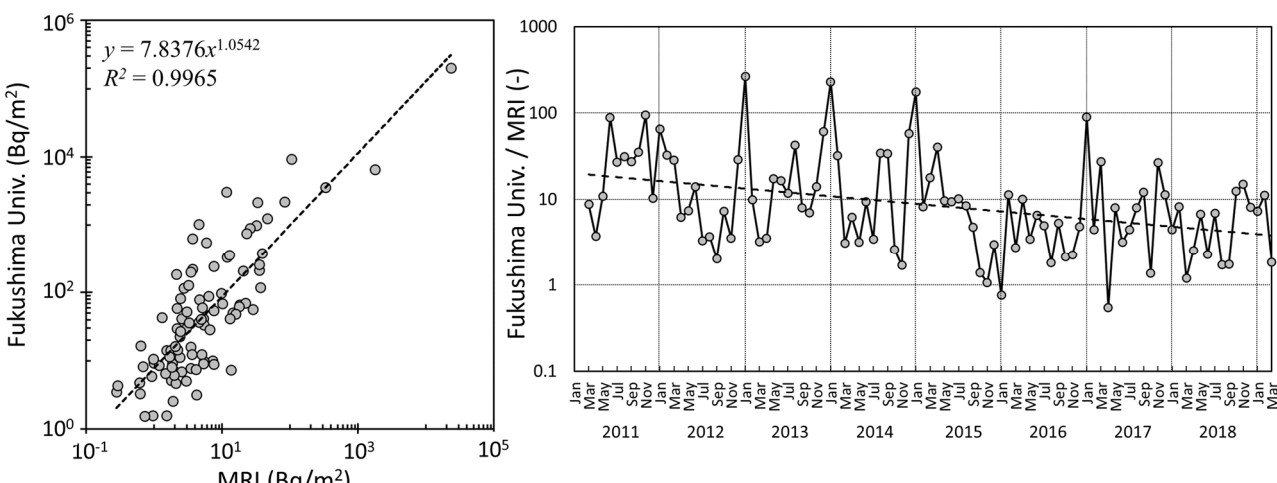

**Figure 10:** (Left) Scatter diagram of the observed surface deposition of $^{137}$Cs at Fukushima University and the MRI from March 2011 to March 2019, with a regression line. (Right) Time series of the ratio of deposition at Fukushima University to deposition at the MRI, with a regression line.



### 3.6 Comparison of deposition amounts at Fukushima University and the MRI

Figure 10 compares the deposition amounts of $^{137}$Cs at Fukushima University (60 km northwest of the FDNPP) and the MRI (170 km southwest of the FDNPP) from March 2011 to March 2019. The deposition data at the MRI are available from Environmental Radioactivity and Radiation in Japan (https://www.kankyo-hoshano.go.jp/data/database/, last accessed: June 14, 2021). There was a significant positive correlation between the deposition amounts of $^{137}$Cs at the two sites. The slope of the regression indicates that the ratio of deposition at Fukushima University to that at the MRI did not change significantly from the initial ratio during the eight years, which is approximately 8-9 times (202200 Bq m$^{-2}$ at Fukushima University and 23100 Bq m$^{-2}$ at the MRI). This indicates that deposition was influenced by emissions from nearby sources and was not substantially influenced by long-range transport at either site. The right panel of Fig. 10 indicates that the deposition ratios at the two sites were approximately 10, with a variation of more than one order of magnitude and peaks in winter (especially January) that decreased slightly over time. The right panel of Fig. 9 shows that the January peak is a feature of Fukushima city and was not observed at the MRI. The possible reasons for the January peak in Fukushima City are discussed later, in Sect. 4.3. The slight decreasing trend was probably due to decontamination, which was ongoing in Fukushima during the period until 2018, as shown later in Table 1. Certainly, natural variations could also have contributed to the decreasing trend.

## 4 Discussion

Even eight years after the FDNPP accident, the surface air activity concentration of $^{137}$Cs had not fallen to the level before the accident, which was at an order of magnitude of $10^{-6}$ Bq m$^{-3}$. In difficult-to-return zones, the surface concentrations sometimes still exceed $10^{-2}$ Bq m$^{-3}$. Based on long-term measurements, this study tries to understand the characteristics of radio-Cs in the air and its deposition and to reveal its origins in order to identify effective ways to reduce radioactivity in contaminated terrestrial ecosystems.

### 4.1 Seasonal variation and possible sources

The current study clearly shows that the surface concentrations of $^{137}$Cs are high from winter to spring, with peaks in March and lows from summer to autumn, in the urban/rural area of Fukushima city (60 km northwest of the FDNPP). It also shows that the deposition amounts of $^{137}$Cs are high in the winter, especially in January, and low from summer to autumn. This seasonal trend is the opposite of that observed in a forested area in the Abukuma Highlands (Tsushima, 30 km northwest of the FDNPP), which was high in the summer (Ochiai et al., 2016; Kinase et al., 2017). From winter to spring, northwesterly winds prevail over the region associated with migrating disturbances, while southeasterly winds prevail over the region associated with the Pacific high. The three simulated monthly mean surface wind fields for January to March and June to August are shown in Kajino et al. (2016).



In summer, Fukushima city is downwind of Tsushima. The surface concentrations of [137]Cs at Tsushima are approximately ten times greater than those at Fukushima city, and Fukushima city is downwind of Tsushima, but there is no enhancement of [137]Cs in summer. The traveling distances of carrier aerosols depend on their aerodynamic diameters. The distance between the two sites is approximately 30 km. The traveling distances of aerosols below < 10 μm are not very different

and are larger than 100 km because their gravitational deposition velocities are negligibly small. On the other hand, the traveling distances rapidly decrease proportionally to a square of the diameter above 10 μm, and the traveling distance of an aerosol with a diameter of 100 μm is an order of 1 km (Kajino et al., 2021). Aerosols below < 10 μm can travel longer distances, but their dry deposition amounts increase significantly from 1 μm to 10 μm (Kajino et al., 2021). Igarashi et al. (2019a) reported that the major proportions of bioaerosols in forests in summer are smaller than 5 μm in diameter and can travel a fairly long

distance. Pollen is much larger than 10 μm, but pollen emission is limited in summer (Igarashi et al., 2019a). Consequently, there was a significant enhancement in surface concentrations in the forests in summer but no enhancement in the downwind urban/rural areas, probably because the carrier aerosols were efficiently deposited onto the ground surface before significant amounts of atmospheric [137]Cs reached the downwind areas. Consistent with our findings presented in Fig. 5, [137]Cs in $PM_c$ was more abundant than that in $PM_f$ and the backup filter particles in summer. To obtain a quantitative understanding of the regional

cycle of atmospheric [137]Cs in the northern part of Fukushima Prefecture, accurate simulations are required in the future.

In winter and spring, the surface concentrations of [137]Cs are probably enhanced due to the local emissions from the nearby sources because the location of the sampling site is upwind of the Abukuma Highlands and the ground surface in the upwind areas of the sampling site in the season (northwest directions) is less contaminated than the site. In winter and spring, [137]Cs in the backup filter particles and $PM_f$ are pronounced in Fukushima city. These characteristics are somewhat different

from those reported in previous studies. Miyamoto et al. (2014) measured the size distributions of radio-Cs with a cascade impactor for two periods, from March 17 to April 1 and May 9 to 13, 2011, at a site 120 km southwest of the FDNPP. They showed that the peak size ranges were 1.2 - 2.1 μm and 0.65 - 1.1 μm in the former and latter periods, respectively. Doi et al. (2013) reported that the peak diameters of the [137]Cs concentration from April 4 to 11 were 1.0 μm and 1.5 μm at Tsukuba, 170 km southwest of the FDNPP. Kaneyasu et al. (2017) measured the size distributions of [137]Cs and other chemical components

six times at Tsukuba from April to September 2011. The peak diameter ranges were 0.49 - 0.7 μm in the earlier stages (before June 9), but the contributions of coarse-mode particles (> 1 μm) increased after June 9, and the second modes appeared in the ranges of 3.5 - 5.2 μm and 7.8 - 11 μm in July and September, respectively. Judging from their measured mass size distributions of Ca, which is assumed to originate from mineral soil, Kaneyasu et al. (2017) concluded that soil particles could be the major carrier of resuspended radio-Cs in Tsukuba. Our XRF analysis indicated that radio-Cs is carried mainly by soil particles in

Fukushima city, but the size distributions are greater in $PM_f$ (<2.5 μm) and in the backup filter particles. If radio-Cs is carried by soil particles, it is natural to presume that the fractions of radio-Cs in $PM_c$ would be large (e.g., Fig. 3 of Kaneyasu et al., 2012 or Fig. 4 of Kaneyasu et al., 2017). One could argue that the bounced coarse mode soil particles are observed in the backup filters, but in fact, the seasonal mean Cs concentrations in the backup filter are positively correlated with $PM_f$ and





negatively correlated with PM$_c$ (Fig. 5). One could further argue that bounced large particles are also collected in PM$_f$ despite the special procedures employed to prevent rebound in the impactor/cyclone system.

There are four possible explanations for these results: (1) If the bouncing effect did not occur in either system, the major sources of radio-Cs in Fukushima city are probably related to combustion (a mass peak below 0.39 μm means that the number peak is approximately 100 nm). If the bouncing effect occurred only in the cascade impactor, (2) if the origin of radio-Cs is soil particles, the size distributions of soil particles in Fukushima city are smaller, or radio-Cs in the soil exists more within finer particles; (3) if the origin of radio-Cs is soil particles, the coarse-mode fraction deposits to the ground surface faster than the fine-mode fraction, such that the proportion of Cs in PM$_f$ is larger in Fukushima city. (4) The bouncing effect occurs in both systems, and the origin of radio-Cs is coarse-mode soil particles. (1) is less likely because there is little chance of the artificial combustion of contaminated biomass. In fact, there were no temporal correlations between the [137]Cs and levoglucosan (a biomass burning marker) surface air concentrations at Tsushima during the forest fire event in the Abukuma Highlands that occurred in March 2013 (Kinase et al., 2018). (3) is also less likely because long-range transport (at least 100 km) is required for the major proportions of coarse-mode particles to deposit to the ground surface, whereas Fukushima city is characterized as the emission source region in that season. In terms of (2), the latter sentence, "radio-Cs in the soil exists more with finer particles", contradicts Kaneyasu et al. (2017), suggesting that the radio-Cs is uniformly distributed on the surface of soil particles. (4) is possible, but rebound is prevented in the impactor/cyclone system, and there is also no evidence that rebound occurred in the impactor/cyclone system (certainly, there is also no evidence that rebound did not occur). Further experiments are required to determine whether (2) or (4) is more likely and whether some sources are missing. As Kaneyasu et al. (2012) and (2017) reported, comparing the size distributions of [137]Cs with those of other chemical components in Fukushima city would be an effective way to investigate the origin of resuspended radio-Cs from winter to spring. Alternatively, a PM$_{2.5}$ cyclone or virtual impactor could be used to separate the fine-mode and coarse-mode particles, that can completely exclude the bouncing effect.

## 4.2 Differences in trends before and after approximately 2015 (natural variation and decontamination)

As described in Sects. 3.1 and 3.2, distinct decrease rates were observed before and after approximately 2015 in both the surface concentrations and the deposition. There may be two main reasons for this: natural variation and decontamination. The natural variation (the dissolved fractions of precipitation or the exchangeable proportions of forest soils discharging faster than other forms from the local ecosystems (Manaka et al., 2016)) was previously described, and the effect of decontamination is presented in some detail here.





**Table 1:** Decontamination achievement ratios in Fukushima City and the surrounding municipalities (Nihonmatsu City, Kawamata Town[a], Date City, and Koori Town)

|  | March 2014 | March 2015 | March 2016 | March 2017 | March 2018 |
|---|---|---|---|---|---|
| Fukushima City |  |  |  |  |  |
| Residential area (Number of houses) | 50.2 % | 62.3 % | 100.0 % | 100.0 % | 100.0 % |
| Public facility (Number of facilities) | 89.3 % | 92.3 % | 97.5 % | 100.0 % | 100.0 % |
| Road (km) | 9.1 % | 16.1 % | 39.6 % | 50.2 % | 100.0 % |
| Agricultural field (ha) | 94.0 % | 94.4 % | 95.2 % | 96.0 % | 100.0 % |
| Forest (living area)[b] (ha) | 5.0 % | 6.3 % | 37.3 % | 80.8 % | 100.0 % |
| The surrounding municipalities |  |  |  |  |  |
| Residential area (Number of houses) | 87.2 % | 97.4 % | 99.6 % | 100.0 % | 100.0 % |
| Public facility (Number of facilities) | 34.3 % | 55.3 % | 80.2 % | 94.9 % | 100.0 % |
| Road (km) | 48.7 % | 56.8 % | 67.5 % | 82.0 % | 100.0 % |
| Agricultural field (ha) | 99.0 % | 99.0 % | 99.6 % | 100.0 % | 100.0 % |
| Forest (living area)[b] (ha) | 23.9 % | 36.7 % | 64.1 % | 88.6 % | 100.0 % |

[a]Only the western part of Kawamata town. The decontamination of areas with an annual cumulative dose exceeding 20 mmSv was assigned to the central government, and that in areas with a dose below 20 mmSv was assigned to municipal governments. The decontamination of the eastern part of Kawamata town was conducted by the central government.

[b]Removal of the litter layer in forests within 20 meters of the forest edge.

Table 1 summarizes the achievement ratios of the scheduled decontamination of different land use types in Fukushima city and the surrounding municipalities (available at https://www.pref.fukushima.lg.jp/site/portal/progress.html, last accessed: June 14, 2021). The municipalities are in the northern part of Fukushima Prefecture, which comprises 55 % forest area, 15 % farmland area, 6 % residential area, and 23 % other areas (https://www.pref.fukushima.lg.jp/uploaded/attachment/42042.pdf, last accessed: June 14, 2021). More than 94 % decontamination was achieved for the farmland area by March 2014. For the residential and public facility areas, some parts were decontaminated by March 2014, but some others were not fully decontaminated until March 2018. For the road and forest areas, decontamination was not completed in most areas by March 2014, but extensive decontamination was conducted from 2014 to 2018. Note that only a part of the forest (20 m from the forest edges) was decontaminated, which accounts for approximately 1 % of the whole forest area of the northern part of Fukushima Prefecture. Additionally, only the litter layer of the forest was removed, and the soil layer remained.

Suppose that if contamination occurred independently of the land use type, approximately 30 % (farmland + half of residential and other) of northern Fukushima was decontaminated by 2014, and an additional 15 % (half of residential and other) was continuously decontaminated by 2018. The difference between the decrease rate from May 2011 to December 2014 (93.1 % $y^{-1}$) and that after (30.7 %$y^{-1}$) was higher than the decontamination rate (30 - 45 % per three to seven years). If the



surface concentration at Fukushima University was affected mainly by the emissions from nearby sources (i.e., within the northern part of Fukushima Prefecture), decontamination would not be the sole reason for the change in the decrease rates before approximately 2015 and after. Natural variation (i.e., regime changes in the chemical forms of radio-Cs) would likely occur during that period. As previously discussed, biotite may have played a key role in the environmental behaviors of radio-Cs in Fukushima city after approximately 2015, but the current study could not identify the key aerosol particles that carried dissolved (or exchangeable) radio-Cs and were abundant in Fukushima city before approximately 2015.

### 4.3 Substantial deposition amounts in January in Fukushima city

The climatological deposition velocities (or the ratios of the deposition rate to the mean surface concentration) in Fukushima City were remarkably high in January 2013, 2014, 2015, and 2017 (Fig. 9). They were approximately one order of magnitude larger than those in the other months. The ratio of the deposition in Fukushima and to that at the MRI was approximately 10 on average, but those in January of those years exceeded 100 (Fig. 10). On the other hand, no peaks were observed in January of 2012, 2016, or 2018.

There are two possible explanations for these results: vertical distribution and the existence of superlarge particles. In terms of the former, the substantial proportions of $^{137}$Cs in the upper air may have caused lower surface concentrations but higher deposition due to the wet removal of $^{137}$Cs aloft. However, due to the northwesterly winter monsoon, the upper air over Fukushima city is also upwind of the Abukuma Highlands; thus, this possibility is less likely. In terms of the latter, superlarge particles (~100 μm or larger in diameter), have settling velocities that are too high (as high as those of drizzle droplets) to enter the high-volume air sampler but that allow them to settle efficiently in a deposition sampler. A similar feature has been observed in the relationship of the deposition and surface concentration of sodium at observation sites near coastal areas (e.g., particles denoted as large sea salt particles (LSPs) in Kajino et al., 2012). The travel distance of such large particles is approximately 1 km (e.g., Kajino et al., 2021), and Fukushima University is surrounded by major roads, such as Route 4 and National Highway E4, within 1 km. January is the month when the highest snow depth occurs in the Fukushima Basin, and the road surface may be wet and muddy due to snow removal work using deicing agents and daytime snow melt on pavements; therefore, road dust emissions from busy transportation activities may be enhanced. The muddy surface conditions may produce even larger road dust particles. Although there is no evidence of the existence of such superlarge particles, they may be a possible reason for the substantial deposition amounts in January in Fukushima city. In fact, substantial amounts of road salt from deicing agents could contribute to roadside PM$_{10}$ samples in winter (Denby et al., 2016), indicating that there could be emissions of particles even larger than 10 μm in diameter. In addition to the direct deposition of $^{137}$Cs to the rain sampler at Fukushima University, the immediate resuspension of deposited $^{137}$Cs associated with road dust or road salt from nearby roads around the university could contribute additional deposition to the rain sampler. Unfortunately, analyses of the surface meteorological observational data for Fukushima City from the JMA, such as temperature, precipitation, snow cover, and wind





speed data, did not reveal the differentiating features between the years with (2013, 2014, 2015, and 2017) and without (2012, 2016, and 2018) high deposition peaks.

## 5 Conclusions

Eight-year measurements of atmospheric $^{134}$Cs and $^{137}$Cs conducted at Fukushima University from March 2011 to March 2019 are summarized in this study. A high-volume sampler, a cascade impactor, and an impactor/cyclone system were used to collect aerosol samples, and the activity concentrations of radio-Cs were detected by high-purity germanium detectors. A precipitation sampler was used to collect deposition samples, and the dissolved and particulate fractions of radio-Cs in the samples were measured. X-ray fluorescence (XRF) analysis was carried out to measure the elemental contents of the aerosol and precipitation samples. The concentration and deposition data measured at Fukushima University were compared with numerical simulation results.

The major findings are itemized as follows:

(1) The observed surface concentrations and deposition at Fukushima University (an urban/rural area of Fukushima city, 60 km northwest of the FDNPP) were high in winter and low in summer; these seasonal trends are the opposite of those observed in a contaminated forest area (Ochiai et al., 2016; Kinase et al., 2018) (30 km northwest of the FDNPP, in the Abukuma Highlands). Resuspension due to bioaerosol emissions (Kinase et al., 2018; Igarashi et al., 2019a) may be substantial in forests but may not be in urban/rural areas. The half-life ($T_h$) and decrease rate ($R_d$) for the eight years were 456 d and 55.6 % y$^{-1}$ for the concentrations and 2.35 y and 29.5 % y$^{-1}$ for the deposition, respectively.

(2) The decreasing trends changed in approximately 2015 and were associated with changes in the dissolved/particulate fractions of $^{137}$Cs in precipitation. The $T_h$ and $R_d$ for concentrations before 2015 were 272 d and 93.1 % y$^{-1}$, whereas they were 825 d and 30.7 % y$^{-1}$ after 2015. The $T_h$ and $R_d$ for deposition before 2015 were 1.10 y and 63.2 % y$^{-1}$, whereas they were 5.39 y and 12.9 % y$^{-1}$ after 2015. The dissolved fractions were higher before 2015, whereas the particulate fractions were higher after 2016. This may have been because the dissolved proportion of radio-Cs discharged faster than its particulate forms from the local terrestrial ecosystems. This is consistent with the findings of Manaka et al. (2016). Decontamination likely also contributed to the difference because the decontamination of some land use types, such as agricultural fields, was completed before 2014, and 100 % of the planned decontamination was completed by March 2018. The contribution of decontamination was estimated in this study to be 30 - 45 % for the three to seven years, which is significantly smaller than the differences in the $R_d$ of the concentrations (93.1 % y$^{-1}$ before 2015 and 30.7 % y$^{-1}$ after 2015). Therefore, decontamination may play a partial role in explaining the differences in $T_h$ and $R_d$ before and after 2015, but changes in the chemical forms of radio-Cs likely play a major role.



(3) The size-resolved measurements showed that the dominant size range in activity in the cascade impactor data was the backup filter (<0.39 μm in diameter, or particles rebounded from larger stages), followed by the 4.2-10.2 μm and >10.2 μm sizes. The backup filter particles were abundant in winter. The seasonal mean $^{137}$Cs concentrations in the backup filter of the cascade impactor were positively correlated with those in the fine-mode aerosols collected by the impactor/cyclone system (PM$_f$) and negatively correlated with those in the coarse-mode aerosols (PM$_c$). PM$_c$ was high in summer. The impactor/cyclone system prevented the bouncing effect, but bouncing may still have occurred during long-duration samplings. The XRF analysis showed that biotite may have played a key role in the environmental circulation of particulate forms of resuspended radio-Cs in Fukushima city after September 2014.

(4) The deposition amounts of $^{137}$Cs in January were remarkably high compared to the surface concentrations of $^{137}$Cs and the deposition amounts of $^{137}$Cs at the MRI. Although we have no observational evidence, we hypothesize that the existence of superlarge particles (~100 μm or larger, with a distance of ~ 1 km or less) associated with snow removal operations on major roads near Fukushima University may be one of the reasons for the remarkable high deposition amounts in January.

Certain issues remained unresolved, and topics for future study are summarized as follows:

(1) The Abukuma Highlands are upwind of Fukushima city in summer. The enhancement of $^{137}$Cs in PM$_c$ in summer is consistent with the fact that most bioaerosols exist in coarse mode. However, if radiocesium is carried mainly by biotite (i.e., soil particles) in winter, there should be an enhancement of $^{137}$Cs in PM$_c$ because major proportions of soil particles exist in coarse mode (e.g., Kaneyasu et al., 2017). On the other hand, sources of Cs-bearing fine-mode particles such as combustion emissions may be less likely. Thus, the main carrier of radio-Cs may be biotite in winter, but this is still not fully confirmed. XRF measurements were conducted for PM$_f$ from September 2014 to January 2018, when the particulate proportions were dominant in the precipitation. Thus, the carrier aerosols of dissolved radio-Cs in Fukushima city are still unknown. As Kaneyasu et al. (2012) and (2017) reported, comparisons of the size distributions of $^{137}$Cs with those of other chemical components in Fukushima city would be an effective way to investigate the origin of resuspended radio-Cs from winter to spring. Alternatively, a PM$_{2.5}$ cyclone or virtual impactor could be applied to separate the fine-mode and coarse-mode particles, which can completely exclude the bouncing effect.

(2) The simulation used in this study was made to be consistent with the surface concentrations in a contaminated forest (Tsushima) and those in a downwind area (the MRI, 170 km southwest of the FDNPP). However, the current study found that the simulated seasonal variation in Fukushima city was totally opposite to the observations. The current study indicated that the deposition velocities applied in the simulation were significantly underestimated. Numerical simulation is a powerful tool for quantitative assessment, but the current simulation requires further improvement. The reasons for the seasonal variations in concentrations and deposition in the different locations need to be investigated with an improved model.



**Data availability**

The observation data used in the study are provided as a Microsoft Excel file in the supplement.

**Author contribution**

AW conducted the long-term measurements together with KN and AS. YN performed the XRF analysis, and MK performed
the numerical simulation. AW, MK, and KN designed the manuscript structure and completed the draft together with all
authors.

**Competing interests**

The authors declare that they have no conflicts of interest.

**Acknowledgments**

This work was supported by Grants-in-Aid for Scientific Research (KAKENHI), grant numbers JP16H01777 and JP24110009.
This work was also supported by the Sumitomo Foundation's Environmental Studies Grant and partly supported by the
Environment Research and Technology Development Fund (JPMEERF20215003) of the Environmental Restoration and
Conservation Agency of Japan. The authors thank Dr. Teruyuki NAKAJIMA of the National Institute for Environmental
Studies and Dr. Haruo TSURUTA of the Remote Sensing Technology Center for proving and installing a high-volume sampler
and Prof. Naohiro YOSHIDA of Tokyo Institute of Technology for the installation of an impactor observation system. The
authors thank Dr. Shogo TOGO of the Isotope Research Center of the University of Tokyo for the radioactivity measurements
of the initial samples and Dr. Yoshiaki YAMAGUCHI, Dr. Zijian ZHANG, Dr. Makoto INAGAKI, Mr. Shunsuke KAKITANI,
Mr. Kazuya FUJIHARA and Mr. Nobufumi FUJITA of Osaka University for their support with the radioactivity measurements
of the air samples. The authors also thank Mr. Kakeru KONNAI of the University of Tsukuba for data visualization, Dr. Akane
SAYA of the MRI for a useful discussion of aerosol deposition processes, and Prof. Tomoaki OKUDA of Keio University for
a useful discussion of the possible artifacts in the impactor measurements.



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
