# Peer review of "Eight-year variations in atmospheric radiocesium in Fukushima city"

_Atmospheric Chemistry and Physics, 2021_

## Referee Comment (RC1)

**General comments**

The authors investigate the role of resuspension in the persistence of airborne radio-cesium in the formerly contaminated city of Fukushima. They focus on dissolved vs particulate fractions of $^{137}$Cs as a supplementary reason to explain the change in the effective half-lives of airborne radio-Cs and its seasonal variation. Airborne concentrations, deposition and size distribution analyses are consistent and relevant. This study adds novelty in the fate of airborne radionuclides and their apparent environmental half-lives.

The height at which some of the used aerosol sampler and impactor were installed may be not perfectly propitious to reveal the exact signature of the resuspension process because this height is too high as compared with ground level where the resuspension process originates. The possible bias could have been investigated. The respective contribution of the fine-mode particle on the airborne concentration may suffer from this particular settlement. I suggest to install a sampler at ground level for a period of one year in parallel with the sampler already installed on the roof of the building to check if the location height has a significant influence on the airborne concenration (what is expected given the vertical profile of aerosol usually observed). This could also be mentioned in the remaining issues to investigate.

It is not clear if large particles have indeed being evidenced by microscopy on the backup filter. It is clear on the other hand that a sampling period as long as 3 weeks may favor particle boucing when using high-volume impactors. Usually, impactor trials last about 2 weeks subject the particle number is very low.

The role and characteristics of biotite are highlighted and the role of the gradual decontamination is scrutinized and show that this sole parameter cannot explain the shift in the half-lives of airborne Cs, thus suggesting the bioavailibility of the different chemical forms of Cs in soils as an important factor.

**Detailed comments**

| Page | Lines | Comment | Example |
|------|-------|---------|---------|
| Abstract | Line 4 | Use « effective half-lives » instead of « half-lives » | |
| | Line 5 | Convert all durations in year and add respectively after | (0.75 and 1.11 yr, respectively) |
| | | Abbreviation for « year » is « yr » | Change it everywhere in the document |
| | Line 6 | I suggest to cut the explanation given line7 and 8 about the shorter half-lives and paste it just after line 5 | |
| | Line 11 | « an evaluation method » instead of « a method of evaluating » | |
| Main text Page 3 | Line 8 | « by precipitation (wet deposition) or during dry weather conditions (dry deposition)» instead of « via precipitation in addition to via dry Deposition » | |
| | Line 11 | Terada et al., 2020). First parenthesis is missing. Remove the period after the final parenthesis | |
| | Line 13 | I suggest to replace « may not be substantial » by « is not expected to be significant » | |
| | Line 20 | since several papers have been published give some references | |

| | | | |
|---|---|---|---|
| | Line 22 | Replace « surface activity concentrations » by « airborne surface concentrations » here and in hereafter in the rest of the document when it refers to concentration of radio-Cs in the atmosphere | Change it everywhere in the document when needed |
| Page 4 | Lines 14, 15 | The reason is just because in Kinase et al. (2018) the air mass did not pass over the observational sites. You cannot let this sentence as it is since it could led to a misinterpretation (i.e. a fire cannot re-emit formerly deposited radio-Cs). Numerous researches performed in the Chernobyl environnement give evidence that fire can re-emit radio-Cs | I would suggest to be very cautious with the role of biomass burning |
| | Line 20 | « (Steinhauser et al., 2015 « instead of « (Steinhouse et al., 2015) » | |
| Page 5 | Line 6 | You can remove the second « located » | |
| | Line 7 | Put the « m » of mountains in captal | Ou Moutain |
| | Line 12 | Replace « where the peaks are » by « where the summits are » | |
| Page 6 | Line 1 | Same remark | |
| | | Replace « at a height of 25 m from» by « at a height of 25 m above» | |
| | Line 19 | « gas-state aerosols » is meaningless. Aerosols are liquid or solid particles. Prefer « volatile or semi-volatile compounds » or « gaseous and volatile or semi-volatile compounds ». Ithink the exact reason of a charcoal cartridge is not for Cs, may be to track possible $^{129}$I revolatilization ? | |
| | Line 6 | I have never heard about « gaseous radioactive cesium ». Cs may be volatized only at temperatures above 650°C but will condense again rapidly as temperature falls. Thus it is considered that it exist only as particle in the atmosphere. | |
| Page 8 | Line 4 | You write « official method ». Is it a national or international method ? Please give a reference | |
| | Line 29 | Instead of « which are usually larger than the submicron size range » you can use « which are usually in the supermicron size range » | |
| Page 9 | Line 13 | Prefer « the decreasing trend » instead of « the decreasing tendency » | |
| | | I suggest « which is much higher that the radioactive hal-life of $^{137}$Cs » instead of « thus, the decrease rate was higher than the rate of radioactive decay of $^{137}$Cs. » | |
| Page 10 | | Because of the numerous data in this plot I strongly suggest to downsize the circles on Figure 2 to see the line between the circles | |
| Page 11 | Line 2 | 202200 should be written 202,200 or 202.2 10$^3$ | |
| | Line 9 | Convert (*Th* = 275 - 756 d) in year. | |
| | Line 9 | « It is tricky… » (or use difficult) instead of « It is hard… » | |
| | Line 18 | Give a reference for the 72.6% | |

| P 16 | Line 2 | Something seems to be missing in « Thus, compared the cascade impactor and the impactor/cyclone measurement data, as shown in Fig. 5. » | |
| | Lines 15-16 | Could you give some evidence of the presence of coarse particles found on the backup filter or explain how you detect them ? | |
| P17 | Line 23 | I do not understand what represents « The eight » in « The eight compositional correlation coefficients » | Introduce the eight compounds before |
| P19 | Line 14 | I suggest « concentrations in Tsushima and Tsukuba (MRI), » instead of « concentrations at Tsushima and the MRI, » | |
| P20 | Line 8 | « factor < 0.1 %. » instead of « factor for < 0.1 %.» | |
| | Line 10 | « From Fig. 9, the value of $b$ for observations is close » instead of « From Fig. 9, $b$ of observation is close » | |
| | Line 11 | The notion of « climatological deposition velocity » which is not conventional should be explained since it differs from what is consensually used as deposition velocity which refers to dry deposition only | |
| P 21 | Line 5 | « overestimation of simulated airborne 137Cs concentration from forests during summer» instead of « overestimation of simulated 137Cs from forests in the summer» | |
| | Line 18 | I think you can be more categorical : which demonstrates the efficacy of wet deposition as compared with dry deposition and which plays … » instead of « but it seems that wet deposition plays… » | |
| | Figure 10 | It would be better to have the same magnitude for the Y-axis and X-axis scales. Currently, at first glance, one could interpret the figure as if deposition at both sites are equal. Please start from $10^{-1}$ to $10^6$ for both axis. | |
| Page 22 | Line 5 | Unless I am misunderstood, I dont agree with « The slope of the regression indicates that the ratio of deposition at Fukushima University to that at the MRI did not change significantly from the initial ratio during the eight years, ». This seems contradictory with what can be seen on fig. 10 (right plot) from where it can be conclude that from the first ratio to the last one there is about a factor of 20 based on the regression line | |
| | Line 7 | 202200 should be written 202,200 or 202.2 $10^{3.}$ the same for 23100 | |
| | Line 7 | « which is approximately 8-9 times higher at the Fukushima University than at the MRI » instead of « which is approximately 8-9 times » | |

| | | | |
|---|---|---|---|
| | Line 9 | Could you please add the average $^{137}$Cs integrated concentration in soils with depth or at the topsoil layer, at both sites | |
| | Line 11 | « January peak is typical at Fukushima city » instead of « January peak is a feature of Fukushima city » | |
| | Line 16 | « the surface air activity concentration of $^{137}$Cs **has** not fallen to the level **prior to** the » instead of «the surface air activity concentration of $^{137}$Cs **had** not fallen to the level **before the**» | |
| | Line 23 | « and low from » instead of « and lows from » | |
| | Line 28 | What is « the Pacific high. » ? | |
| P 23 | Line 2 | « and Fukushima city is downwind of Tsushima, » is already mentioned line 1 | |
| | Line 7 | I do not see the interest to mention the case of aerosol with a such a high diameter since they are exceptionnally detected or correspond to very specific activities or at coastal sites. Withour refering to such extrem value, it could be more interesting to give an example of more « common » aerosol sizeslike 20 or 30 µm even if again they remain much less abundant than 10 µm | |
| | Line 12 | Aside from the diameter which is sensitive to gravitational deposition, the efficient deposition onto the ground can be attributed to rain deposition. While dry deposition is almost permanent this suggest that wet deposition is also more or less regular if not permanent (this cannot be seen based on the precipitation amount which is on a monthly basis | |
| P 24 | Line 5 | « If the bouncing effect occurred only in the cascade impactor, » The place of this sentence seems strange. Does it already correspond to the second possible explanation ? Isf so the « 2) » should be placed before the sentence | |
| | Line 3 to 10 | The reading is not straightforward and the text would gain to be more intelligible. | |
| | Line 13 - 30 | The suggestion of an enhanced dust emission during snow period (even if it does have an effect given the short distance between the sampling location and the roads) would worth to be investigated before asserting. May be this idea could be developed in another paper.

After line 13, I would suggest to shift to line 30 starting with « Unfortunately, analyses of the surface meteorological observational data for Fukushima City from the JMA, such as temperature, precipitation… » | In order to keep with what has been measured and what can be interpreted with a relative high confidence. I would skip this snow section because it is too uncertain |
| P27 | Line 15 | Convert 456 d in year | |

| | Line 18 | « changed approximately in 2015 » or « changed around 2015 » instead of « changed in approximately 2015 » | |
|---|---|---|---|
| | Line 19-20 | Convert 272 d and 825 d in year | |
| | Line 23 | In the conclusion, no need to repeat « This is consistent with the findings of Manaka et al. (2016). » | |
| P28 | Lines 9-12 | I would shift this item in the remaining unresolved issues if not deleted (see my previous comment about snow and mud | |

---

## Author Comment (AC1)

Dear anonymous referee #1,

We are very grateful for your detailed review and constructive comments and your time for RC1. Thanks to your review, our manuscript has been substantially improved, especially for details and preciseness. We have considered all your comments in the revised manuscript.

Point-by-point responses to your comments are written in blue in this letter.

With best regards,
Akira WATANABE, Mizuo KAJINO, and Kazuhiko NINOMIYA

General comments:

[1] The authors investigate the role of resuspension in the persistence of airborne radio-cesium in the formerly contaminated city of Fukushima. They focus on dissolved vs particulate fractions of $^{137}$Cs as a supplementary reason to explain the change in the effective half-lives of airborne radio-Cs and its seasonal variation. Airborne concentrations, deposition and size distribution analyses are consistent and relevant. This study adds novelty in the fate of airborne radionuclides and their apparent environmental half-lives.
[1] Thank you for your evaluation.

[2] The height at which some of the used aerosol sampler and impactor were installed may be not perfectly propitious to reveal the exact signature of the resuspension process because this height is too high as compared with ground level where the resuspension process originates. The possible bias could have been investigated. The respective contribution of the fine-mode particle on the airborne concentration may suffer from this particular settlement. I suggest to install a sampler at ground level for a period of one year in parallel with the sampler already installed on the roof of the building to check if the location height has a significant influence on the airborne concenration (what is expected given the vertical profile of aerosol usually observed). This could also be mentioned in the remaining issues to investigate.
[2] Thank you for your good suggestion. We fully agree with your point. We inserted the following statement in the first item of the remained issue:

"The height of our measurement (building roof) is higher than the other measurements referenced in this study (near the ground). When the observation site is characterized as an emission source, there should be a clear vertical difference in concentration, and thus the concentration measured at Fukushima University is not equivalently comparable with the other location data. It may be comparable when the site is characterized as a downwind region, because turbulent mixing during transport may reduce the vertical difference. In the future, parallel sampling near the ground and rooftop will need to be installed to characterize the sampling locations and to quantify the vertical differences at the Fukushima University site."

[3] It is not clear if large particles have indeed being evidenced by microscopy on the backup filter. It is clear on the other hand that a sampling period as long as 3 weeks may favor particle boucing when using high-volume impactors. Usually, impactor trials last about 2 weeks subject the particle number is very low.

[3] There is no evidence that the rebound occurred in the backup filter in our samples, as written in Sect. 4.2. The electron microscopy revealed evidence of rebound of soil particles and bioaerosols in the same experimental setup (Kinase et al., 2018) as explained in Sect. 2.2.3.

Considering together with the comment #2 of RC3, we added the following statement as item (2) in the remained issue section of Conclusion:

"(2) The rebound issue of the impactor and the cyclone/impactor instruments have not yet been resolved. Parallel sampling is also required for the size-resolved measurements using normal filters and filters with adhesive materials such as vacuum grease. The additional microscopy of the filters is even more useful."

[4] The role and characteristics of biotite are highlighted and the role of the gradual decontamination is scrutinized and show that this sole parameter cannot explain the shift in the half-lives of airborne Cs, thus suggesting the bioavailibility of the different chemical forms of Cs in soils as an important factor.

[4] We agree with your point. We inserted the following statement at the beginning of the second item of major findings in Conclusion:

"The bioavailability of different chemical forms of radio-Cs in soils may be an important factor to determine the tendencies of concentrations and deposition at Fukushima University."

[5] Detailed comments:
[5] Replies are inserted in the tables.

| Page | Lines | Comment/Reply | Example |
|---|---|---|---|
| Abstract | Line 4 | Use « effective half-lives » instead of « half-lives »
We changed it accordingly. | |
| | Line 5 | Convert all durations in year and add respectively after
We changed it. Also 756d → 2.07 and 4.69 yr. We also changed them in the figures. | (0.75 and 1.11 yr, respectively) |
| | | Abbreviation for « year » is « yr »
We changed them all accordingly. | Change it everywhere in the document |
| | Line 6 | I suggest to cut the explanation given line7 and 8 about the shorter half-lives and paste it just after line 5
We changed it accordingly. | |
| | Line 11 | « an evaluation method » instead of « a method of evaluating »
We changed it accordingly. | |

| Main text Page 3 | Line 8 | « by precipitation (wet deposition) or during dry weather conditions (dry deposition)» instead of « via precipitation in addition to via dry Deposition »
We changed it accordingly. | |
|---|---|---|---|
| | Line 11 | Terada et al., 2020). First parenthesis is missing. Remove the period after the final parenthesis
We changed it accordingly. | |
| | Line 13 | I suggest to replace « may not be substantial » by « is not expected to be significant »
We changed it accordingly. | |
| | Line 20 | since several papers have been published give some references
Thank you. The following sentences are the references so we added the phrase « as follows » at the end of the sentence. | |
| | Line 22 | Replace « surface activity concentrations » by « airborne surface concentrations » here and in hereafter in the rest of the document when it refers to concentration of radio-Cs in the atmosphere
We changed them all. | Change it everywhere in the document when needed |
| Page 4 | Lines 14, 15 | The reason is just because in Kinase et al. (2018) the air mass did not pass over the observational sites. You cannot let this sentence as it is since it could led to a misinterpretation (i.e. a fire cannot re-emit formerly deposited radio-Cs). Numerous researches performed in the Chernobyl environnement give evidence that fire can re-emit radio-Cs
Yes, we agree with you. They/we didn't have any evidence that forest fire in Fukushima did not reemit Cs. We changed the relevant sentences as follows :
« Certainly, it is not indicated that the forest fire did not reemit radio-Cs, because in fact wildfire played a key role in the migration of radio-Cs in the Chernobyl case » | I would suggest to be very cautious with the role of biomass burning |
| | Line 20 | « (Steinhauser et al., 2015 « instead of « (Steinhouse et al., 2015) »
Thank you. We changed it accordingly. | |
| Page 5 | Line 6 | You can remove the second « located »
We changed it accordingly. | |
| | Line 7 | Put the « m » of mountains in captal
We changed it accordingly. | Ou Moutain |
| | Line 12 | Replace « where the peaks are » by « where the summits are »
We changed it accordingly. | |
| Page 6 | Line 1 | Same remark
We changed it accordingly. | |

| | | Replace « at a height of 25 m from» by « at a height of 25 m above»
We changed it accordingly. | |
|---|---|---|---|
| | Line 19 | « gas-state aerosols » is meaningless. Aerosols are liquid or solid particles. Prefer « volatile or semi- volatile compounds » or « gaseous and volatile or semi-volatile compounds ». I think the exact reason of a charcoal cartridge is not for Cs, may be to track possible [129]I revolatilization ?
Yes. We changed it to « volatile or semi-volatile compounds ». | |
| Page 7 | Line 6 | I have never heard about « gaseous radioactive cesium ». Cs may be volatized only at temperatures above 650°C but will condense again rapidly as temperature falls. Thus it is considered that it exist only as particle in the atmosphere.
Yes, it is already obvious. Thus, the whole sentence was removed. | |
| Page 8 | Line 4 | You write « official method ». Is it a national or international method ? Please give a reference
It is the national method developed based on international literatures. We inserted the reference as « MEXT, 1976 », which is available at https://www.kankyo-hoshano.go.jp/wp-content/uploads/2020/12/No3.pdf (in Japanese) | |
| | Line 29 | Instead of « which are usually larger than the submicron size range » you can use « which are usually in the supermicron size range »
We changed it accordingly. | |
| Page 9 | Line 13 | Prefer « the decreasing trend » instead of « the decreasing tendency »
We changed it and all relevant places. | |
| | | I suggest « which is much higher that the radioactive hal-life of [137]Cs » instead of « thus, the decrease rate was higher than the rate of radioactive decay of [137]Cs. »
Thank you for the suggestion. Please note that the relevant sentence was removed according to the #7 of RC2. | |
| Page 10 | | Because of the numerous data in this plot I strongly suggest to downsize the circles on Figure 2 to see the line between the circles
We downsized them by two points, from 6 to 4. | |
| Page 11 | Line 2 | 202200 should be written 202,200 or 202.2 $10^3$
We changed it to 202 $10^3$. (Please also refer to #3 of RC3). | |
| | Line 9 | Convert ($T_h$ = 275 - 756 d) in year.
We changed it to ($T_h$ = 0.753 – 2.07 yr) | |

| | Line 9 | « It is tricky » (or use difficult) instead of « It is hard »
 We changed it to "tricky". | |
| --- | --- | --- | --- |
| | Line 18 | Give a reference for the 72.6%
 We inserted "(Fig. 3)" here. | |
| P 16 | Line 2 | Something seems to be missing in « Thus, compared the cascade impactor and the impactor/cyclone measurement data, as shown in Fig. 5. »
 We simply changed the whole sentence to « Thus, the cascade impactor and the impactor/cyclone measurement data are compared in Fig. 5. » | |
| | Lines 15-16 | Could you give some evidence of the presence of coarse particles found on the backup filter or explain how you detect them ?
 We did not find any evidence for this. We suspect the bouncing effect might occur because the cascade impactor with the similar experimental setup (Kinase et al., 2018) showed substantial amounts of soil dust and bioaerosol particles on the backup filter as found by the electron microscopy observation. But, yes, the observation locations are different, so there is no evidence that the bouncing occurred in the Fukushima city case, except the current statistical analysis. Thus, we changed the relevant sentence from
 « It appears that bouncing effect occurred »
 to
 « It appears that bouncing effect might occur ».

 Please also refered to our reply to your RC1 #3 – item (2) in the remained issue section of Conclusion. | |
| P17 | Line 23 | I do not understand what represents « The eight » in « The eight compositional correlation coefficients »
 We inserted the following phrase
 (the common 10 species shown in Fig. 7 minus the two, $SiO_2$ and $Al_2O_3$) | Introduce the eight compounds before |
| P19 | Line 14 | I suggest « concentrations in Tsushima and Tsukuba (MRI), » instead of « concentrations at Tsushima and the MRI, »
 We changed it to "Tsushima and Tsukuba". | |
| P20 | Line 8 | « factor < 0.1 %. » instead of « factor for < 0.1 %.»
 We changed it accordingly. | |
| | Line 10 | « From Fig. 9, the value of *b* for observations is close » instead of « From Fig. 9, *b* of observation is close »
 We changed it accordingly. | |
| | Line 11 | The notion of « climatological deposition velocity » which is not conventional should be explained since it | |

| | | differs from what is consensually used as deposition velocity which refers to dry deposition only. | |
|---|---|---|---|
| | | We inserted the following statement just after the relevant sentence as follows :
« Note that the concept of climatological deposition velocity differs from that of dry deposition velocity; the dry deposition velocity is defined as the ratio of the mass flux divided by the concentrations, but this climatological deposition velocity is the ratio of total (dry plus wet) deposition amounts devided by concentrations without the concept of mass flux. To account for the wet deposition flux, both in-cloud and below-cloud concentrations are needed, but such vertical measurement data is not available. ». | |
| P 21 | Line 5 | « overestimation of simulated airborne $^{137}$Cs concentration from forests during summer» instead of « overestimation of simulated $^{137}$Cs from forests in the summer»
We changed it accordingly. | |
| | Line 18 | I think you can be more categorical : which demonstrates the efficacy of wet deposition as compared with dry deposition and which plays … » instead of « but it seems that wet deposition plays… »
We changed the whole sentences as follows :
« As mentioned above, the magnitude of the instant deposition velocity and our climatological deposition velocity are not directly comparable, but it demonstrates the efficacy of wet deposition as compared with dry deposition. Wet deposiiton plays an important role in the removal of resuspended $^{137}$Cs-bearing particles from the air. » | |
| | Figure 10 | It would be better to have the same magnitude for the Y-axis and X-axis scales. Currently, at first glance, one could interpret the figure as if deposition at both sites are equal. Please start from $10^{-1}$ to $10^6$ for both axis.
We changed it accordingly. | |
| Page 22 | Line 5 | Unless I am misunderstood, I dont agree with « The slope of the regression indicates that the ratio of deposition at Fukushima University to that at the MRI did not change significantly from the initial ratio during the eight years, ». This seems contradictory with what can be seen on fig. 10 (right plot) from where it can be conclude that from the first ratio to the last one there is about a factor of 20 based on the regression line
We agree with you. There is a slight trend in the right panel of Fig. 1, Fukushima deposition drops somewhat | |

| | | faster than Tusukuba deposition, with strong seasonal variations. We simply removed the relevant sentences and changed the whole sentences as follows :
 « There was a significant positive correlation between the deposition amounts of $^{137}$Cs at the two sites, but the ratios varied substantially over time. The right panels of Fig. 10 indicates that the deposition ratios at the two sites were approximately 10, which is almost the same level as the initial amounts (202 10$^3$ Bq m$^{-2}$ at Fukushima University and 17.6 10$^3$ Bq m$^{-2}$ at the MRI), with a variation of more than one order of magnitude and peaks in winter (especially January) that decreased slgithly over time. » | |
| | Line 7 | 202200 should be written 202,200 or 202.2 10$^{3.}$ the same for 23100
 We changed to the latter, to be consistent with the previous modification. Also, please note that the value 23100 was incorrect (the old undetermined value was used) and it was changed to 17.6 10$^3$ Bq m$^{-2}$. | |
| | Line 7 | « which is approximately 8-9 times higher at the Fukushima University than at the MRI » instead of « which is approximately 8-9 times »
 The relevant sentence was removed. | |
| | Line 9 | Could you please add the average $^{137}$Cs integrated concentration in soils with depth or at the topsoil layer, at both sites
 The relevant sentence was removed. | |
| | Line 11 | « January peak is typical at Fukushima city » instead of « January peak is a feature of Fukushima city »
 We changed it accordingly. | |
| | Line 16 | « the surface air activity concentration of $^{137}$Cs **has** not fallen to the level **prior to** the » instead of «the surface air activity concentration of $^{137}$Cs **had** not fallen to the level **before the**»
 We changed it accordingly. | |
| | Line 23 | « and low from » instead of « and lows from »
 We changed it accordingly. | |
| | Line 28 | What is « the Pacific high. » ?
 "pressure system" was inserted at the end. | |
| P 23 | Line 2 | « and Fukushima city is downwind of Tsushima, » is already mentioned line 1
 We removed the latter part. | |
| | Line 7 | I do not see the interest to mention the case of aerosol with a such a high diameter since they are exceptionnally detected or correspond to very specific activities or at coastal sites. Without refering to such | |

| | | | |
|---|---|---|---|
| | | extrem value, it could be more interesting to give an example of more « common » aerosol sizes like 20 or 30 µm even if again they remain much less abundant than 10 µm

Thank you for your suggestion. We changed the sentence as follows :
« the traveling distance of an aerosol with a diameter of several 10 µm is an order of 10 km. » | |
| | Line 12 | Aside from the diameter which is sensitive to gravitational deposition, the efficient deposition onto the ground can be attributed to rain deposition. While dry deposition is almost permanent, this suggest that wet deposition is also more or less regular if not permanent (this cannot be seen based on the precipitation amount which is on a monthly basis)

Yes, it is a good point. We changed the relevant sentence as follows :
« Consequently, there was a significant enhancement in concentrations in the forests in summer but no enhancement in the downwind urban/rural areas, probably because the carrier aerosols were efficiently depositied onto the ground surface « by wet deposition in addition to dry deposition » before significant amounts of atmospheric $^{137}$Cs reached the downwind areas. | |
| P 24 | Line 5 | « If the bouncing effect occurred only in the cascade impactor, » The place of this sentence seems strange. Does it already correspond to the second possible explanation ? Isf so the « 2) » should be placed before the sentence

Yes, it was awkward. By considering the next comment (on Line 3 to 10) together, we reorganized the whole sentences as follows :
(1) If the bouncing effect did not occur in either system, the major sources of radio-Cs in Fukushima city are probably related to combustion (a mass peak below 0.39 µm means that the number peak is approximately 100 nm). (2, 3) If the bouncing effect occurred only in the cascade impactor, (2) the size distributions of soil particles in Fukushima city are smaller, or radio-Cs in the soil exists more within finer particles; or (3) the coarse-mode fraction deposits to the ground surface faster than the fine-mode fraction, such that the proportion of Cs in PMf is larger in Fukushima city. (4) The bouncing effect occurs in both systems, and the origin of radio-Cs is coarse-mode soil particles. | |

| | | | |
|---|---|---|---|
| | Line 3 to 10 | The reading is not straightforward and the text would gain to be more intelligible.
Please see our reply to your previous comment. | |
| | Line 13 - 30 | The suggestion of an enhanced dust emission during snow period (even if it does have an effect given the short distance between the sampling location and the roads) would worth to be investigated before asserting. May be this idea could be developped in another paper. After line 13, I would suggest to shift to line 30 starting with « Unfortunately, analyses of the surface meteorological observational data for Fukushima City from the JMA, such as temperature, precipitation… »
We agree with your suggestion. It is discussed in our follow-up paper and thus the whole sentences are removed here. | In order to keep with what has been measured and what can be interpreted with a relative high confidence. I would skip this snow section because it is too uncertain |
| P27 | Line 15 | Convert 456 d in year
We changed to 1.25 yr. (and changed to 1.24 yr after recalculation according to #6 of RC3.) | |
| | Line 18 | « changed approximately in 2015 » or « changed around 2015 » instead of « changed in approximately 2015 »
We changed it to "around 2015". | |
| | Line 19-20 | Convert 272 d and 825 d in year
We changed them to 0.745 and 2.26 yr, respectively. (then to 0.754 and 2.07 after recalculation according to #6 of RC3.) | |
| | Line 23 | In the conclusion, no need to repeat « This is consistent with the findings of Manaka et al. (2016). »
We removed the whole sentence. | |
| P28 | Lines 9-12 | I would shift this item in the remaining unresolved issues if not deleted (see my previous comment about snow and mud)
We shifted this to the #1 of the remaining issue. We are designing a field experiment somehow to prove this effect. Since we removed the « snow and mud » part according to your previous comment, this item is simplified as follows :
« The deposition amounts of [137]Cs in January at Fukushima University were remarkably high compared to the concentrations of [137]Cs and the deposition amounts of [137]Cs at the MRI. The reason needs to be investigated in the future. » | |

---

## Author Comment (AC2)

Dear anonymous referee #2,

We are very grateful for your time, acceptance of review, and very fine and constructive comments for RC2. Thanks to your review, our manuscript has been substantially improved, especially for your suggestion on the period of change the in tendencies. We have considered all your comments in the revised manuscript accordingly.

Point-by-point responses to your comments are written in blue in this letter.

With best regards,
Akira WATANABE, Mizuo KAJINO, and Kazuhiko NINOMIYA
* * *
[General comments]

[1] This paper has studied and contributed to not only the better understanding for the chemistry of Cs in the atmosphere but also the earlier rebirth from nuclear accident. This study is quite challenging to the non-reproducible event based on the eight-year measurement. Therefore, this study should include many uncertainties. Under such a difficult situation, this paper can give scientists many useful information and knowledge including unsolved agenda. In this meaning, this study should be appropriate for publishing in ACP.
[1] Thank you very much for your evaluation.

On the other hand, I expect the authors to describe and give suggestions to readers for the points below.
- [2] Give us a clearer scientific (physical, chemical, mathematical or some other) reason why the authors consider that the year of "2015", neither "2014" nor "2016", was the turning point in time series, especially for Fig. 3.
- [2] Thanks to your comment, we could find the mathematical reason, indicating that something happened in 2015. In the previous manuscript, it could be either 2014 or 2016 and thus we used the fuzzy term "approximately 2015" which can be from 2014 to 2016. We additionally calculated the half-lives before and after 2014 and 2016, and as you pointed, revealed that there was a significant difference for the 2016 case of the deposition (Fig. 3).

    In addition, after considering the #6 of RC3, we excluded the initial stage data which contains the effects of primary emission in addition to resuspension from the tendency calculation. Thus, please be aware that all the tendency values are changed accordingly.

    The ratios of half-lives before to after January 2014, January 2015, January 2016 are 2.75, 3.06, 2.84, (all around 3.0) respectively for concentrations, but those for depositions are 3.61, 3.64, and 8.12. Some drastic change might occur within the year of 2015.

We made a new figure showing changes in the half-lives of depositions before and after a particular date from 2014 to 2016 as below.

[Figure]

Unfortunately, because half-lives after 2015 varied substantially depending on the start month (it exceeded 100 year for some cases), the ratio after August 2015 is really unstable. Therefore, we gave up to show this figure in the main text but it was inserted in the supplementary data in the excel file (please find tab number #7).

Anyway, this figure indicates that there is not sudden but gradual change started to occur from spring to summer of 2015.

- [3] Why should the fractions in dissolved and particulate change suddenly in 2015? Give us scientific reasons/comments/discussions in detail more.
- [3] We couldn't find any specific reason for this in addition to what we extensively discussed: "dissolved Cs discharged faster from environment than undissolved". Still, however, as shown above (our reply to #2 of your comment RC2), we could find that there was not a sudden change, but gradual changes occurred during 2015.

  Consequently, by taking #2 and #3 of RC2 into account, the following changes are made in the manuscript:

(1) A new paragraph is made as the 3rd paragraph of Sect. 3.1 as follows:

"The regression analysis is also performed over different time periods, but the results are not substantially different. The $T_h$ and $R_d$ before December 2013, 2014, and 2015 are 0.670, 0.753, and 0.900 yr, and 103, 92.0, and 77.0 % $yr^{-1}$, respectively. The $T_h$ and $R_d$ after January 2014, 2015, and 2016 are 2.05, 20.7, and 2.56 yr, and 33.8, 33.5, and 27.1 % $yr^{-1}$, respectively."

(2) A new paragraph is made as the 4th paragraph of Sect. 3.2 as follows:

"The regression analysis is also applied over different time periods and we found a remarkable change in 2015. The Th before December 2013, 2014, and 2015 are similar 1.09, 1.30, and 1.56 yr, respectively, but Th after January 2014, 2015, and 2016 are 3.98, 4.69, and 12.67 yr, respectively. The ratios of half-lives (after to before) of the three periods are 3.64, 3.61, and 8.12, respectively, indicating that there could be a remarkable change in the tendency between January 2015 and January 2016. Time series of changes in the ratio before and after a particular date from 2014 to 2016 are illustrated in Fig. S1. Due to the lack of data numbers, the half lives after 2015 varied substantially depending on the start month (it exceeded 100 yr for some cases). However, it is obvious that the ratio is stable before January 2015 around four and start to increase from the spring to summer of 2015. We may be able to conclude that the regime change in the physicochemical properties of radio-Cs occurred during the year of 2015."

- [4] When we compare the results between forest sites and current study sites, the sampling height above the ground level might be different. Is there any influences on the measurement results and the subsequent interpretations of the data? (Around p.10, LL.12-14)
- [4] Please also refer to our reply to the comment #2 of RC1. The vertical difference matters for emission source but not in downwind areas. Whether the Fukushima University site is emission or downwind may differ by seasons and relative abundance of carrier aerosols (for example, downwind for forest aerosols and emission source for soil particles (especially for road dust)). These statements are itemized in the remained issue section of Conclusion as follows:

"The height of our measurement (building roof) is higher than the other measurements referenced in this study (near the ground). When the observation site is characterized as an emission source, there should be a clear vertical difference in concentration, and thus the concentration measured at Fukushima university is not equivalently comparable with the other location data. It may be comparable when the site is characterized as a downwind region, because turbulent mixing during transport may reduce the vertical difference. In the future, parallel sampling near the ground and rooftop will need to be installed to characterize the sampling locations and to quantify the vertical differences at the site."

[Specific comments]

[5] p.3, LL.11-12, Why do we need bracket; (The……2011).
[5] We removed the bracket.

[6] p.6, L.13,    Is "Shibata" "Sibata"?
[6] Thank you for the good point. We did not realize it.

[7] p.9, L.14,    Why do the authors select power of X, not exponential function?
[7] The function of power of X fitted better than the exponential function, but it was contradictory with the latter statement saying that "This demonstrates that the concentration decreased "exponentially". We simply removed the sentences because we have already discussed the trend elsewhere in the manuscript. Same for the first paragraph of Sect. 3.2.

[8] p.16, LL.11-13,    On the description of "We ………and autumn.", is the reason for "We can assume" either the present measurement results or other references? If the former is, is the description of "our measurement strongly indicates" better than "we can assume"?
[8] Thank you. We modified the sentences as follows:
"There was a negative correlation between $PM_c$ and $PM_f$, which strongly indicates that… and autumn."

[9] p.17, L.17, Why can the author simply say "the samples have similar origins" although the precipitation is influence by not only below cloud scavenging but also in cloud scavenging?
[9] Yes, it was an overstatement. We changed the whole sentence as follows:
"Composition differences are not very remarkable, the correlation coefficients for the compositions among samples are above 0.9."

[10] p.20, L.13 and p.27, LL.28-29,    Is there any conflicts between two sentences of "it seems that wet deposition plays an important role in the removal of…" and "Therefore, decontamination may play a partial role in explaining the differences…"?
[10] Thank you for your question. It is not contradicting because the former sentence describes the removal mechanism of resuspended $^{137}$Cs in the air, whereas the latter sentence describes the reasons in changes in the trend before and after 2015.

To be clearer, the following change is made for the former statement as follows:

_Former place:_
From "it seems that wet deposition plays an important role in the removal of resuspended $^{137}$Cs-bearing atmospheric aerosols"

To "it seems that wet deposition plays an important role in the removal of resuspended $^{137}$Cs-bearing particles from the air"

---

## Author Comment (AC3)

Dear anonymous referee #3,

We are very grateful for your time, acceptance of review, and very fine and constructive comments for RC3. Thanks to your review, our manuscript has been substantially improved. Specifically, the comment of removing initial period from the resuspension tendency is very meaningful. We have considered all your comments in the revised manuscript accordingly.

Point-by-point responses to your comments are written in blue in this letter.

With best regards,
Akira WATANABE, Mizuo KAJINO, and Kazuhiko NINOMIYA

[General comments]

[1] This paper observed valuable data with long and steady efforts and showed important and new knowledge (such as change of solubility of Cs containing aerosols in deposition) which can be useful in environmental radioactivity science and various atmospheric science. They aimed to clarify sources of Cs containing aerosol particles, their activities in the environment, and their future estimations. This paper can contribute to the understanding of the environmental cycle of aerosols such as aerosol production, transporting, and deposition by using Cs as a tracer. In addition, this paper rose an important suggestion for the model improvements through improvements of aerosol deposition estimations. Therefore, I think this paper is appropriate for publishing from ACP.
[1] Thank you very much for your evaluation.

[2] However, this paper remains a large uncertainty to the aerosol size measurements. They are making great efforts to evaluate the performance of the 6-stage impactor with the cyclone/impactor instrument. They discussed Cs containing aerosols on the backup filter using a large part of this paper, however, the bouncing effects of large particles had not been denied. Rather, significant bouncing effects on both instruments were shown, but there was no evidence to deny bouncing effects. Therefore, I think a large contribution of bouncing effects will be quietly significant. These results and discussions about the particle sizes make this paper confusing. If authors suggest that the significant contributions of fine aerosol particles, more accumulation of reliable and accurate evidence for this point should be required (such as parallel observations using the same impactors with normal filters and adhesive material applied filter (such as vacuum grease), microscopic observation, and others).
[2] Thank you for raising the very important issue. As for the parallel sampling, we made a new item in the remained issue section of Conclusion for the vertical measurement according to the comments of RC1(#2) and RC2 (#4). In the similar manner, we made an additional item (2) in the remained issue section of Conclusion as follows:

"(2) The rebound issue of the impactor and the cyclone/impactor instruments have not

yet been resolved. Parallel sampling is also required for the size-resolved measurements using normal filters and filters with adhesive materials such as vacuum grease. The additional microscopy of the filters is even more useful."

Other common comments are below.

[3] Please check the significant figures (such as P8 L6) and make the numbers easier to see.
[3] Thank you for your comments. We changed it from 1.158 to 1.16 here. As the significant figures of observation data are from 2 to 3 (please see the supplement excel file), we determined the significant figures of all values derived from statistical analyses not exceeding 3. Thus, we carefully checked all values in other locations and changed them as follows:
"202200 Bq m$^{-2}$" → "202 10$^3$ Bq m$^{-2}$"
"48232X$^{-1.944}$" → Whole sentence removed (Please refer to #7 of RC2)
"0.678" → "0.68" for readability
"7.8376x$^{1.0542}$" → "7.84x$^{1.05}$"
"$R^2$=0.9965" → "$R^2$=0.997"

[4] Please clarify the relationship between river sediments and this paper more. This observation does not seem to contribute to the paper significant
[4] In the revised manuscript, the relationship between river sediments and this paper is elaborated in Sect. 2.4 (the methodology section), when it is first appeared in the manuscript as follows:
"River sediments that characterize the surface soils of the Nakadori valley were also measured to assess the composition correlations with the airborne and deposition samples."

[5] "surface air concentration", "atmospheric radioactivity concentration", and "surface concentration" were confused (atmospheric radioactivity concentration?).
[5] Thank you very much for your comment. After considering #5 of RC1 regarding Line 22 of Page 3, we changed "surface activity concentrations" to "airborne surface concentrations". To keep the consistency in the manuscript, we only used the two terms "airborne surface concentrations" and simply "concentrations" throughout the manuscript.

[6] This paper calculated radioactivity decreases using data from 2011 (the early stage after the accident). In the early stage, resuspension is not dominant. It is necessary to distinguish periods of primary stages and resuspension stages. This point will affect the results of future estimations and the rate of Cs discharges.
[6] Thank you very much. It is a very good point. We re-calculated all tendency factors by excluding the initial period of data (March and April 2011 of deposition).

For the concentration measurement (Fig. 2), it started on May 18, 2011, we used all data and so the values have not been changed. For the monthly deposition measurement (Fig. 3), March 2011 data should be excluded, but it is a question of

whether to exclude the April data. Based on Katata et al. (2015), total emission amounts of $^{134}Cs+^{137}Cs$ from FDNPP in April was 0.7 PBq, which accounted for 2.4% of total emission (28.7 PBq). The value "2.4%" seems to be small but with compared to the resuspension rate (less than 1%/yr; Kajino et al., ACP, in press; doi:10.5194/acp-2021-687), it may not be negligible. Thus, we decided to exclude the April data, too.

[7] Please check again for the referencing.
[7] Do you mean the consistency of references in the main text and in the reference list? If so, we changed from "Steinhouse et al., 2015" to "Steinhouser et al., 2015" as found by RC1 and "Kinase et al., 2017" to "Kinase et al., 2018" as found by you (#16 of RC3), but we could not find any further inconsistencies. If it is not the case, kindly please specify the errors. Thank you very much for your time.

[Specific comments]

[8] P6 L15: "The range of particle sizes…" can be shortened as "The 50% cut of particle sizes…". Then, "(Note…" can be deleted.
[8] We changed it accordingly.

[9] P7 L5: Were carbon filters used for gaseous Cs analysis? Isn't the purpose such as measuring iodine?
[9] Yes, it is. By taking #5 of RC1 (regarding Line 19 of Page 6 and Line 6 of Page 7) into account, we modified the sentences.

[10] P7 L12: Coarse mode samples were collected on the quartz fiber filter. This point is inconsistent with the L8 paragraph.
[10] XRF is applied only for the fine mode fractions which were collected in the glass bottles. The relevant sentence is modified as follows:
"Aerosols larger than 2.5 µm were collected on quartz fiber filters in the system and thus only the fine mode particles in the glass bottles were measured by XRF."

[11] P13 L14: Also impactor sampling is "time-resolted measurement".
[11] Yes, it is. We changed it to "all size observation as presented in Fig. 2".

[12] P13 L24: 2016 and 2017? Or 2015 DJF and 2015 DJF? Also, the opposite trend can be seen in 2013.
[12] I am sorry for the confusion. The sentence is the explanation of only >10.2 µm and not the explanation of comparison between >10.2 µm and backup filter. You mean the opposite trend in 2013 is for >10.2 µm and backup filter, but we meant that the trend of >10.2 µm of 2016 and 2017 (high in JJA) are different (not opposite) from other years (high in DJF and MAM9.

We modified the sentences from
"The seasonal variations in the largest particle fraction, larger than 10.2 µm, are interesting. The trend appears to be synchronized with that of the backup filter particles (high in DJF and MAM), but the opposite trend was observed in 2016 and 2017 (high in

JJA)."

to
"The seasonal variations in the largest particle fraction, larger than 10.2 µm, are interesting: high in DJF and MAM (same as the backup filter) but high in JJA in 2016 and 2017."

[13] P13 L25: Is the seasonal trend of 1.3-2.1 µm particles significant? It looks quietly stable.
[13] Yes, it is not significant. These parts are discussing the trends of minor size ranges so the following sentences are all REMOVED:
"The contributions of other fractions, i.e., 0.49-4.2 µm, were small in the measured period. Even though the contributions were small, the seasonal trend of 0.39-0.69 µm was similar to that of the backup filter particles, but that of 1.3-2.1 µm was similar to that of 4.2-10.2 µm."

[14] P15 L3: Too long sampling intervals (recommended operating time is up to 24 hours). Were some parallel observations using the same impacter instruments with and without oil? Were some microscopic checks or any other checks had done?
[14] Unfortunately, no. Since it is the very important and critical point, we itemized the issue in the remained issue part of Conclusion as follows:
"The rebound issue of the impactor and the cyclone/impactor instruments have not yet been resolved. Parallel sampling is also required for the size-resolved measurements using normal filters and filters with adhesive materials such as vacuum grease. The additional microscopy of the filters is even more useful."

[15] P22 L: Is the wording "difficult -to-return zone" is right? I could find "Areas where it is expected that the residents have difficulties in returning for a long time" in the Japanese governmental report.
[15] Yes, it is found for example in the national report issued by Government of Japan (p.33). (last accessed: December 8, 2021)
https://www.iaea.org/sites/default/files/japan_2nd_em_2012.pdf
The term is currently used in official web pages as well, for example,
https://www.pref.fukushima.lg.jp/site/portal-english/en03-08.html

[16] P22 L26: 2017? 2018?
[16] Thank you. We changed it to 2018.

[17] P24 L17: Authors showed a paper (Okuda et al., 2015) as a reference indicating the rebounds of large particles using the impactor/cyclone instrument with long sampling periods in P7.
[17] We modified the relevant sentence as follows:
"(4) is possible because Okuda et al. (2015) showed that the long-duration impactor/cyclone measurement could be associated with the bouncing effect despite the use of silicone oil.

[18] P27 L13: At the city site, some references showed the same results of seasonal variations (such as Kitayama et al., 2016; Kinase et al., 2019).
[18] We inserted the following underlined sentences in the relevant place:
"these seasonal trends are the same of those observed in the city area (Kitayama et al., 2016) and the opposite of those observed in a contaminated forest area (Ochiai et al., 2016; Kinase et al. 2018)".

We could not find the paper "Kinase et al. (2019)" of either Dr. Takeshi Kinase and Dr. Sakae Kinase. We suppose it as Kinase et al. (2018), because their measurement was conducted in the forest area, but they also claimed that the trend in the forest differed from that in the other locations. We already cited their paper in the sentence so we additionally cited only Kitayama et al. (2016) for the Fukushima city case.

In case if we misunderstand the reference, kindly please provide the full information of the relevant paper. Thank you for your time.

[19] P28 L3: As mentioned above, these results include high risks of misunderstandings about Cs containing particle sizes.
[19] We agree with you in this point. We removed all size values from the item #3 and modified it as follows:

"(3) The size-resolved measurements revealed that seasonal variations of $^{137}$Cs of different sizes are different from each other. Due to the possible bouncing effect of the cascade impactor and long-duration measurement of the impactor/cyclone system, it is hard to quantify the values, but the current measurement indicates that the dominant particles and their sizes may be distinct depending on the season. The XRF analysis showed that biotite may have played a key role in the environmental circulation of particulate forms of resuspended radio-Cs in Fukushima city after September 2014."